# ALS mutations disrupt self-association between the ubiquilin STI1 hydrophobic groove and internal placeholder sequences

Joan Onwunma[1,4], Saeed Binsabaan [2,4], Shawn P Allen [1,2,4], Sachini R Thanthirige [1,2], Deepika Gaur [1,2], Banumathi Sankaran[3] & Matthew L Wohlever [1,2]✉

## Abstract

Ubiquilins are molecular chaperones that play multifaceted roles in proteostasis, with point mutations in UBQLN2 leading to altered phase-separation properties and amyotrophic lateral sclerosis (ALS). Our mechanistic understanding of this essential process has been hindered by a lack of structural information on the STI1 domain, which is essential for ubiquilin chaperone activity and phase separation. Here, we present the first crystal structure of a ubiquilin-family STI1 domain bound to a transmembrane domain (TMD), and show that ALS mutations disrupt the STI1-TMD interaction. We further demonstrate that ubiquilins contain multiple conserved internal sequences that bind to the STI1 domain, including the PXX-repeat region that is a hotspot for ALS mutations. We propose that these placeholder sequences prevent solvent exposure of the STI1 hydrophobic groove and contribute to the multivalency that drives ubiquilin phase-separation. Together, this work provides a new paradigm for understanding how STI1 domains modulate ubiquilin chaperone activity and phase separation, and offers insights into the molecular basis of ALS pathogenesis.

**Keywords** Ubiquilins; Proteostasis; Biomolecular Condensates; Membrane Proteins; Amyotrophic Lateral Sclerosis
**Subject Categories** Post-translational Modifications & Proteolysis; Structural Biology; Translation & Protein Quality

## Introduction

Amyotrophic lateral sclerosis (ALS) is a fatal neurodegenerative disease characterized by the progressive degeneration of motor neurons (Le et al, 2016). A hallmark of ALS is dysregulated protein homeostasis, resulting in altered dynamics, composition, and material properties of biomolecular condensates (Portz et al, 2021; Lau et al, 2018; Renaud et al, 2019; Conicella et al, 2016;

Rajendran and Castañeda, 2025). The ubiquilin family of proteins are key players in the cellular proteostasis network, facilitating the degradation of misfolded proteins, chaperoning uninserted membrane proteins, and interacting with other cellular components to regulate phase separation (Zheng et al, 2020; Ma et al, 2023; Black et al, 2023; Whiteley et al, 2017; Alexander et al, 2018). Humans have three widely expressed ubiquilin paralogs, UBQLN1, UBQLN2, and UBQLN4, which are linked to numerous neurodegenerative diseases, with point mutations in UBQLN2 causing dominant, X-linked Amyotrophic Lateral Sclerosis (ALS) (Deng et al, 2011; Edens et al, 2017; Fahed et al, 2014; Bertram et al, 2005). Despite decades of research, the mechanistic basis for how these mutations disrupt ubiquilin function and lead to disease is unclear.

A defining feature of ubiquilins is a propensity for phase separation. While the physiological role of phase separation remains opaque, growing evidence links ALS mutations with altered phase separation properties (Dao et al, 2019). For example, histology studies of ALS patients and mouse models of neurodegenerative diseases show increased formation of UBQLN2 puncta (Safren et al, 2024; Gerson et al, 2021; Thumbadoo et al, 2024). Furthermore, in vitro studies show that many ubiquilins have altered material properties (Riley et al, 2021; Dao et al, 2022). Finally, recent reports demonstrate that UBQLN2 condensates can promote fibril formation of α-synuclein (Takei et al, 2025). Understanding how ALS mutations alter ubiquilin phase separation properties remains an important goal for the field.

Ubiquilins are characterized by an N-terminal Ubiquitin Like (UBL) domain, two STI1 domains, and a C-terminal Ubiquitin Associated (UBA) domain, separated by long stretches of intrinsically disordered regions (IDRs) (Fig. 1A) (Zheng et al, 2020). Yeast has a single ubiquilin homolog, known as Dsk2. While the domain organization is largely conserved between Dsk2 and ubiquilins, one notable difference is that Dsk2 has a single STI1 domain. UBQLN2 is unique among ubiquilin paralogs as it contains a PXX repeat region, which is a hotspot for many ALS mutations. Recent studies have identified additional ALS mutations in both UBQLN2 STI1 domains, which remain poorly characterized (Safren et al, 2024). Interestingly, the STI1-II domain is

[1]Department of Chemistry & Biochemistry, University of Toledo, Toledo, OH, USA. [2]Department of Cell Biology, University of Pittsburgh, Pittsburgh, PA, USA. [3]Lawrence Berkeley National Lab, Berkeley Center for Structural Biology, Molecular Biophysics and Integrated Bioimaging, Berkeley, CA, USA. [4]These authors contributed equally: Joan Onwunma, Saeed Binsabaan, Shawn P Allen. ✉E-mail: wohlever@pitt.edu

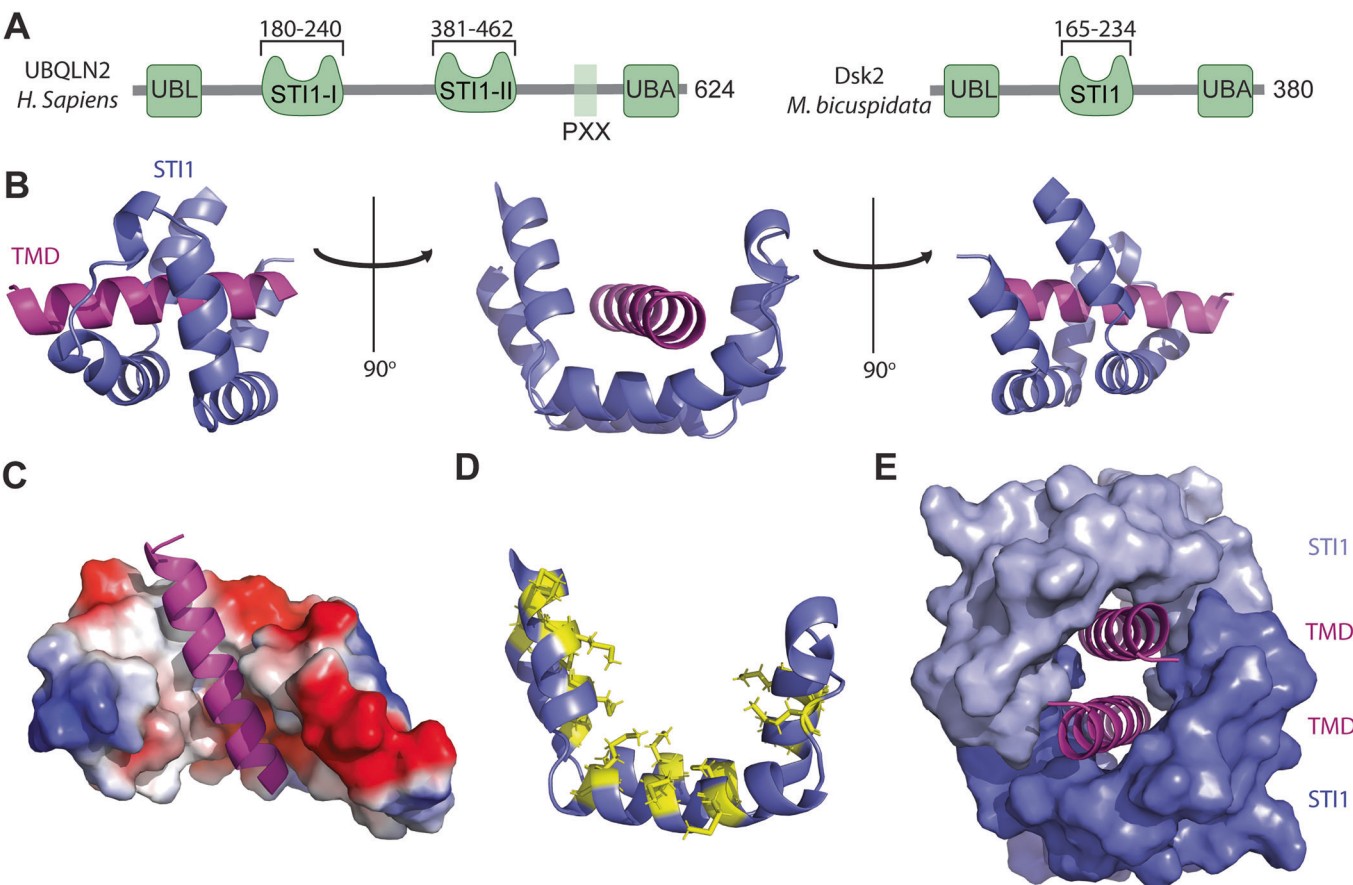

**Figure 1. Crystal structure of the Dsk2 STI1 domain bound to a TMD.**

(A) Domain diagram of human UBQLN2 and *M. bicuspidata* Dsk2 with STI1 domain boundaries annotated. The crystallization construct includes residues 165–234 of *M. bicuspidata* Dsk2. (B) Cartoon representation of a single Dsk2 STI1 domain with 90° rotations. The TMD is colored magenta, and the STI1 domain is colored slate. FrzS domains omitted for clarity (PDB: 9CKX). (C) Surface representation of the Dsk2 STI1 domain colored by electrostatics; blue is positive, red is negative, and white is hydrophobic. (D) The Dsk2 STI1 hydrophobic groove is enriched for methionine residues, which are shown as yellow sticks. (E) Dimerization of two Dsk2 STI1 domains forms a hydrophobic cavity that fully encapsulates two TMDs. Individual subunits are shown as a surface, and TMDs are shown as cartoons. FrzS domains omitted for clarity.

essential for UBQLN2 condensate formation, but how this domain contributes to phase separation is unclear (Dao et al, 2018).

Despite an initial characterization as proteasomal shuttle factors that facilitate degradation of ubiquitinated client proteins (Hjerpe et al, 2016; Kleijnen et al, 2000; Walters et al, 2002; Dantuma et al, 2009), ubiquilins also have intrinsic chaperone activity, which is comparatively understudied (Stieren et al, 2011; Kurlawala et al, 2017). Previous work has shown that ubiquilin STI1 domains can directly bind to moderately hydrophobic transmembrane domains (TMDs), a property that is particularly enriched among mitochondrial membrane proteins (Itakura et al, 2016; Rao et al, 2016). This original analysis was carried out on substrates with single-pass TMDs that, while exclusively composed of hydrophobic residues, varied in the ratio of highly vs. moderately hydrophobic residues within the TMD. While the STI1 domains are rich in methionine residues, a hallmark of many TMD-binding proteins (Guna and Hegde, 2018; Pleiner et al, 2023; Stefer et al, 2011), the molecular basis for selectively binding moderately hydrophobic TMDs is unclear. STI1 domains are a common motif in proteostasis and can

be broadly categorized as either co-chaperones or adapters of the Ubiquitin Proteasome System (Fry et al, 2021a; Lin et al, 2021). There is little structural data on the proteasome adapter family STI1 domains, which include ubiquilins. While several modeling studies, including AlphaFold, have predicted the rough shape of the ubiquilin STI1 domain (Fry et al, 2021a; Jumper et al, 2021), these predictions have low confidence. The lack of experimental structural data on STI1 domain chaperone activity is thus a major roadblock in the field.

Here, we present the first crystal structure of the ubiquilin family STI1 domain bound to a TMD. Structure-function analysis shows that ALS-causing mutations disrupt the STI1 hydrophobic groove and TMD binding. We also developed a novel barcoded binding assay to discover that human ubiquilin STI1 domains preferentially bind to hydrophobic sequences with low helicity, a property shared by multiple sequences within ubiquilins. We then demonstrate that ubiquilins contain multiple conserved, internal sequences that bind to the STI1 hydrophobic groove, which we term placeholder sequences. Importantly, these placeholder

sequences include the PXX repeat motif in UBQLN2, which is a hotspot for ALS mutations that modulate ubiquilin phase separation. We propose that the STI1-placeholder interaction prevents solvent exposure of the STI1 hydrophobic groove and contributes to the multivalency that drives ubiquilin phase separation, thereby explaining why ALS-causing mutations alter phase separation properties.

# Results

## Crystal structure of the STI1 domain bound to a TMD

Ubiquilin STI1 domains are flexible and have short-lived interactions with substrates, making them challenging targets for traditional structural biology approaches (Itakura et al, 2016; Fry et al, 2021a). To stabilize the STI1-TMD interaction, we generated a fusion construct consisting of the STI1 domain from *M. bicuspidata* Dsk2, a variant of the VAMP2 TMD, and the receiver domain from *Myxococcus xanthus* social motility protein FrzS as a crystallization scaffold (Fraser et al, 2007). FrzS was chosen because it robustly crystallizes at high resolution and has the N and C-termini positioned to allow for optimal interaction between the TMD and STI1 domain (Maurer et al, 2023). The resulting construct formed crystals within three days at 4 °C. We solved the crystal structure via molecular replacement with the FrzS domain with an overall resolution of 1.98 Å (Table 1; Appendix Fig. S1A).

The STI1 domain is composed of 6 α-helices that assemble into a helical-hand or U-shape. Each of the sides of the helical-hand contains two α-helices connected by short loops with ~90° turns (Fig. 1B). The STI1 domain forms a hydrophobic groove enriched in methionine residues (Fig. 1C,D), a common feature of many membrane protein chaperones (Hegde and Keenan, 2022; Guna and Hegde, 2018). The overall shape is roughly similar to the AlphaFold 3 prediction; however, there are numerous differences in the details of TMD packing with the hydrophobic groove, highlighting the importance of experimental structural biology approaches (Appendix Fig. S1B). The TMD binds the bottom of the hydrophobic groove and buries ~2970 Å² of hydrophobic surface area with each STI1 monomer. Key residues in the hydrophobic groove that contact the TMD include L190, M193, A207, M210, L211, M214, M218, M225, M228, and M229, which correspond to residues 30, 33, 47, 50, 51, 54, 58, 65, 68, and 69 in the crystallization construct (Appendix Fig. S1C).

Surprisingly, the asymmetric unit shows two STI1 domains stacked head-to-head to create a fully enclosed hydrophobic cavity capable of chaperoning two TMDs (Fig. 1E). The presence of multiple TMDs within the hydrophobic groove is reminiscent of the crystal structure of the TMD chaperone and targeting factor GET3 (Mateja et al, 2015). The dimer has a domain swap, with the TMD of subunit 1 binding to the STI1 domain of subunit 2 (Appendix Fig. S2A). The dimerization of the STI1 domain is appealing for several reasons. First, the STI1-II domain is essential for UBQLN2 dimerization (Dao et al, 2024). Second, the formation of a hydrophobic cavity by the dimer ensures that no part of the hydrophobic TMD is solvent-exposed.

To further probe the significance of the dimer, we performed size-exclusion chromatography and observed that the crystallization construct runs as an oligomer (Appendix Fig. S2B).

**Table 1. Data collection and refinement statistics**

|  | STI1-Frzs-TMD |
| --- | --- |
| **Data collection** |  |
| Space group | P 1 21 1 |
| Cell dimensions |  |
| a, b, c (Å) | 67.41 63.92 67.43 |
| α, β, γ (°) | 90.0, 98.83, 90.0 |
| Resolution (Å) | 66.63–1.98 (2.09–1.98) |
| Unique reflections | 36912 (4887) |
| $\langle I / \sigma I \rangle$ | 17.7 (2.1) |
| $R_{merge}$[a] | 0.056 (0.827) |
| $R_{pim}$ | 0.024 (0.367) |
| Completeness (%) | 93.3 (85.9) |
| Redundancy | 6.4 (6.0) |
| **Refinement** |  |
| Resolution (Å) | 46.127–1.98 |
|  | (2.03–1.98) |
| $R_{work}$[b]/$R_{free}$[c] (%) | 22.00– 23.21 |
| Number of atoms | 3516 |
| Protein (non-H) | 3392 |
| Water | 124 |
| Avg B-factors (Å²) | 55.45 |
| Protein | 55.67 |
| Water | 49.65 |
| R.m.s. deviations |  |
| Bond lengths (Å) | 0.005 |
| Bond angles (°) | 0.80 |
| Ramachandran |  |
| Favored, Allowed, Outliers (%) | 99.09, 0.91, 0.0 |
| Clashscore | 6 |

Values in parentheses represent the outer resolution shell.
[a]$R_{merge} = (|(\Sigma I - \langle I \rangle)|)/ (\Sigma I)$, where $\langle I \rangle$ is average intensity of multiple measurements.
[b]$R_{work} = \Sigma_{hkl} ||F_o(hkl)|| - F_c (hkl)||/\Sigma_{hkl} |F_o(hkl)|$.
[c]$R_{free}$ represents the cross-validation R factor for 5% of the reflections against which the model was refined.

Interestingly, the crystallization construct runs with an apparent molecular weight of ~75 kDa, which corresponds to a trimer, although this may be due to the extended conformation of the crystal construct. Dimer formation is unlikely to be an artifact of crystallization, as the two FrzS domains do not contact each other in the asymmetric unit (Appendix Fig. S2C).

Analysis of the crystal structure shows that the majority of the dimer interface arises from the contacts between the two TMDs. To test the contribution of the TMDs to dimer formation, we generated a version of our crystal construct lacking the TMD (STI1-FrzS-ΔTMD) and performed size-exclusion chromatography. This construct eluted significantly later, indicating that the TMD is a major contributor to the oligomerization of the crystal construct. The apparent molecular weight of the STI1-FrzS-ΔTMD construct was 45 kDa, which corresponds to a dimer (Appendix Fig. S2B).

To test if the STI1-FrzS-ΔTMD construct forms a stable dimer, we sought to disrupt STI1-STI1 contacts. We identified residues M29, N37, and M71 as key regulators of the STI1-STI1 dimer interface in our crystal structure (Appendix Fig. S2D). These residues correspond to amino acids M189, N197, and M231 in the *M. biscuspidata* Dsk2. We generated the M29D, N37A, M71D triple point mutation in the STI1-FrzS-ΔTMD construct and performed size-exclusion chromatography. We observed a modest shift in elution volume, but the construct still did not elute at the volume expected for a monomer (Appendix Fig. S2B). The most parsimonious explanation for these results is that the crystallization construct runs anomalously large on size-exclusion chromatography due to an extended conformation. We conclude that the STI1 domain forms a hydrophobic groove to chaperone substrates and that the STI1-TMD interaction is a major driver of oligomerization in the crystal construct.

## ALS mutations are predicted to disrupt the STI1 hydrophobic groove

To gain a better understanding of how ALS mutations in the STI1 domain affect ubiquilin function, we performed a sequence alignment of the *M. biscuspidata* STI1 domain against the STI1-I or STI1-II domains of human UBQLN1, 2, and 4 (Appendix Fig. S3A,B). The resulting sequence alignment was then mapped onto our crystal structure, and the position of UBQLN2 ALS mutations was analyzed. Our structure predicts that the Q425R and M446R mutations will disrupt the hydrophobic groove via insertion of a charged residue (Fig. 2A–C). In the crystal structure, these residues form hydrophobic interactions with the TMD (Appendix Fig. S3C,D). Notably, the AlphaFold prediction of UBQLN2 yields similar results (Appendix Fig. S3E,F).

The P189T and P440L mutations also occur in the UBQLN2 STI1 domains but are not predicted to contact the TMD (Fig. 2D,E). Our analysis shows that the P189T and P440L mutations occur in short loops with a ~90° turn between the bottom and side of the helical hand. The prolines and preceding residues have Phi-Psi angles that are favored by Proline (Richardson and Richardson, 1988). We hypothesized that mutation of these highly conserved Proline residues would disrupt the folding and/or dynamics of the STI1 domain.

To test these predictions, we used the PURExpress In Vitro Transcription/Translation (IVT) system to produce $^{35}$S labeled substrates in a detergent-free environment (Shimizu et al, 2001). The PURExpress reaction was supplemented with physiological concentrations (3 μM) of recombinantly purified FLAG-tagged UBQLN2 variants. As the PURExpress system contains no other chaperones, membrane, or detergent, binding of the membrane protein to ubiquilins can be assessed by a simple anti-FLAG immunoprecipitation (IP). We chose the mitochondrial membrane protein Omp25 as our model substrate, as it had been previously shown to bind to UBQLN1 STI1 domains (Itakura et al, 2016).

We observed that the P189T mutation led to a ~90% decrease in Omp25 binding, whereas the P440L or M446R mutations had no effect (Fig. 2F,G). This was an unexpected result given the high sequence similarity between the STI1-I and STI1-II domains. Size-exclusion chromatography analysis shows that the P189T mutant behaves identically to WT UBQLN2 (Appendix Fig. S4A). Further

control reactions show that all UBQLN2 variants have equal IP efficiency under our assay conditions (Appendix Fig. S4B).

Having ruled out trivial explanations, we came up with two models to explain our results: (1) Omp25 binds exclusively to the STI1-I domain or (2) Omp25 binds to both STI1 domains, but has higher affinity for the STI1-I domain and the concentrations of Omp25 and/or UBQLN2 are such that Omp25 binds predominantly to the STI1-I domain under our assay conditions. To differentiate between these models, we repeated the binding assay with 23.5 μM UBQLN2 instead of 3 μM, with the expectation that the higher concentration of UBQLN2 would allow Omp25 to bind to the STI1-II domain. Consistent with our hypothesis, we observed that Omp25 now binds to both WT and P189T UBQLN2 (Appendix Fig. S4C,D).

We then asked if combining the P189T mutation in the STI1-I domain with either the Q425R, P440L, or M446R mutation in the STI1-II domain would further disrupt Omp25 binding. We were unable to purify UBQLN2 double mutants at a sufficient concentration to perform the in vitro binding assay with 23.5 μM UBQLN2. We therefore co-transfected HEK cells with HA-tagged UBQLN2 variants and FLAG-tagged Omp25 under the control of the strong CMV promoter and performed a co-IP experiment. As expected for overexpression conditions, the P189T mutant behaved similarly to WT UBQLN2 (Fig. 2H). However, combining the P189T mutation in the STI1-I domain with either Q425R, M446R, or P440L mutation in the STI1-II domain leads to substantial defects in Omp25 binding (Fig. 2I). Consistent with our crystal structure, the P189T/P440L double mutant had the most severe effect, whereas the P189T/Q425R model had the mildest effect. We conclude that substrates bind to both STI1 domains, that the STI1-I domain has a higher affinity for substrate binding, and that ALS-causing mutations likely inhibit substrate binding by disrupting the structure and/or dynamics of the STI1 hydrophobic groove.

## Development of a barcoded binding assay to study substrate binding to the STI1 domain

It was previously proposed that ubiquilins selectively bind to moderately hydrophobic TMDs (Itakura et al, 2016). Our crystal structure shows a highly hydrophobic groove, which cannot easily explain a preference for binding moderately hydrophobic TMDs. We therefore decided to rigorously test this model by developing a competitive binding assay.

Substrate was again produced via the PURExpress IVT system supplemented with physiological concentrations of recombinantly purified FLAG-tagged ubiquilin. To better mimic the complexity of the cellular environment, we sought to simultaneously present ubiquilins with multiple closely related substrates. We developed a pool of six model substrates based on the TMD of Omp25, a mitochondrial tail-anchored (TA) protein and known ubiquilin substrate (Itakura et al, 2016; Guna et al, 2022). To assess TMD hydrophobicity, we used the Grand Average of Hydrophobicity (GRAVY) score. This simple metric reflects the average hydropathy of an amino acid sequence using the Kyte-Doolittle scale, with larger positive numbers indicating a more hydrophobic sequence (Kyte and Doolittle, 1982). We then systematically varied the TMD GRAVY score by mutating residues to Alanine or Leucine. To differentiate substrates by SDS PAGE, a variable number of SH3 domains were added to serve as a "barcode" (Fig. 3A). The SH3

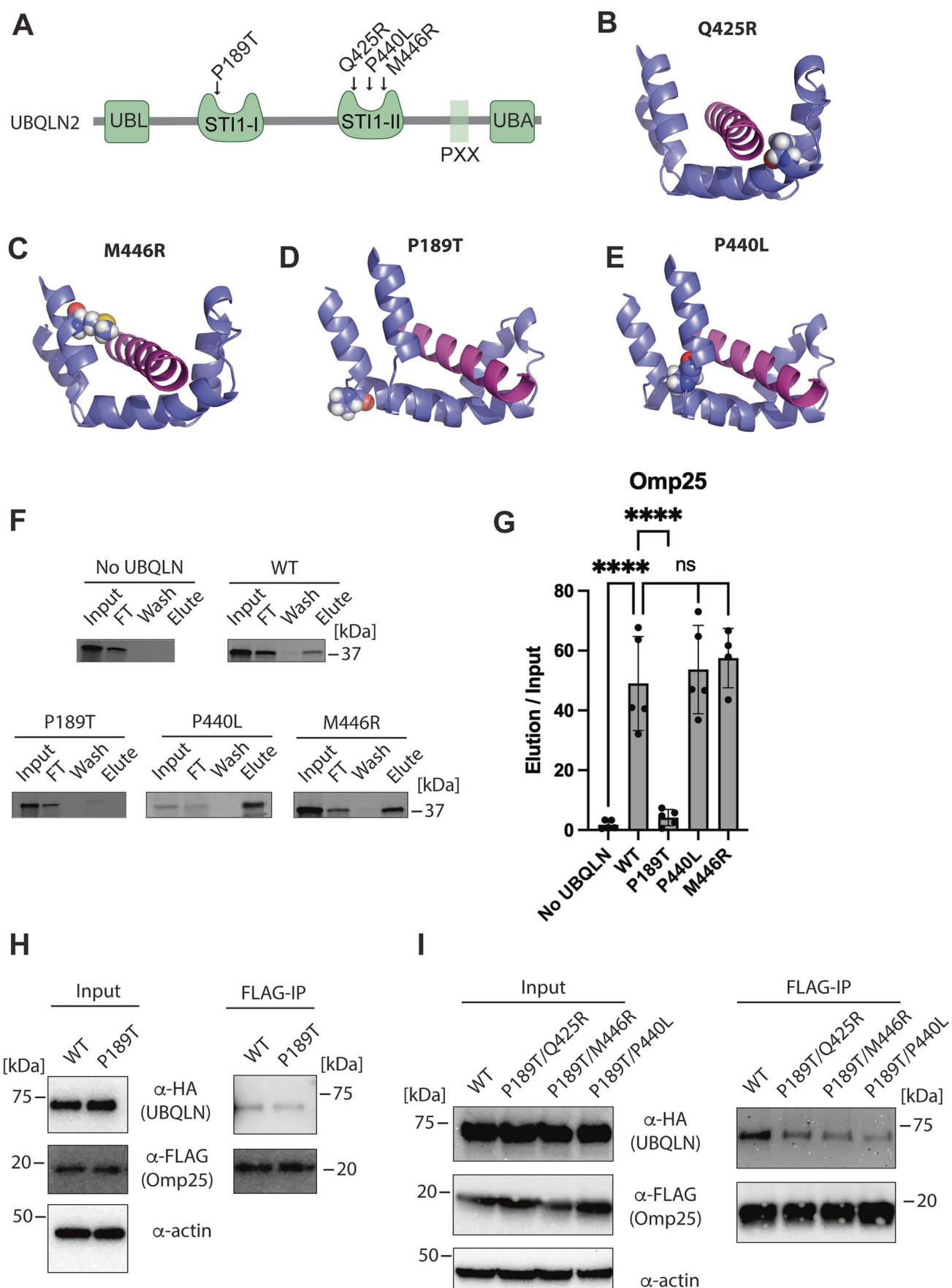

**Figure 2.  UBQLN2 ALS mutations disrupt the STI1 hydrophobic groove.**

(A) Domain diagram showing ALS-causing mutations located in the UBQLN2 STI1 domains. (B–E) The UBQLN2 sequence was mapped onto the Dsk2 structure, and the location of ALS-causing mutations was analyzed. The UBQLN2 Q425R (B) and M446R (C) mutations are predicted to insert a charged residue into the STI1 hydrophobic groove. Residues P189 (D) and P440 (E) form the ~90° turns between α-helices in the helical hand motif. Mutation of these highly conserved residues is predicted to disrupt formation and/or dynamics of the hydrophobic groove. (F) The P189T mutation disrupts Omp25 binding to UBQLN2. The $^{35}$S labeled Omp25 TMD was produced by PURExpress IVT in the presence of 3 µM recombinant 3xFLAG-UBQLN2 variants, and binding was assayed by anti-FLAG immunoprecipitation. Ubiquilins were eluted with 3xFLAG peptide. Representative western blots from $N \geq 4$. (G) Quantification of data in (F), error bars are standard deviation, centered on the mean for $N \geq 4$. $P$ values were calculated by one-way ANOVA with Dunnett's post hoc test; ****$P < 0.0001$, ns: not significant. Exact $P$ values: No UBQLN $= 4.8 \times 10^{-6}$, P189T $= 9.9 \times 10^{-6}$, P440L $= 0.90$, and M446R $= 0.60$. (H) P189T UBQLN2 shows only a modest defect in Omp25 binding when overexpressed in HEK cells. HA-Tagged UBQLN2 variants and FLAG-tagged Omp25 were co-transfected into HEK cells, and binding interaction was monitored by DSP cross-linking followed by anti-FLAG immunoprecipitation. Western blot is representative of two independent experiments. (I) Combining the P189T mutation in the STI1-I domain with either Q425R, M446R, or P440L mutation in the STI1-II domain leads to defects in Omp25 binding in HEK cells. The experiment was performed as described in (H). Western blot is representative of two independent experiments. Source data are available online for this figure.

domain was chosen because it is small, well-folded, and contains no methionine residues, which would bias the $^{35}$S signal intensity (Kurochkina and Guha, 2013). After testing the expression of each plasmid individually, the six plasmids for a substrate series were mixed together such that the expression of each model substrate was roughly equal. Ubiquilin-substrate complexes were purified by anti-FLAG IP, and binding was calculated by comparing the amount of substrate in the elution and input fractions.

Control experiments show that binding is dependent on the STI1-TMD interaction. Deletion of the TMD led to minimal substrate in the elution fraction, arguing that there is no interaction between ubiquilins and the SH3 domains (Fig. 3B). Similarly, the previously characterized RRR TMD mutant (Itakura et al, 2016), which has three Arginine mutations in the middle of the TMD, disrupts substrate binding (Fig. 3C). Finally, we used our crystal structure and AlphaFold model to generate the UBQLN1 binding mutant, which contains two mutations in each STI1 domain that are predicted to disrupt substrate binding. We observed no binding of any of the Omp25 substrates to the binding mutant, confirming that binding is dependent on the STI1-TMD interaction (Appendix Fig. S5A).

To test if the number of SH3 domains influences interaction with ubiquilins, we developed a substrate series where all six substrates have the WT Omp25 TMD (Appendix Fig. S5B). As expected, there is no statistically significant difference in substrate binding (Appendix Fig. S5C). We were concerned that binding assays with UBQLN2 could be influenced by the interaction of the proline-rich PXX repeat region with the SH3 domains. To test for this, we generated a UBQLN2 construct lacking the PXX repeat region and tested for binding to Omp25 (Appendix Fig. S5D). We observed no significant differences between WT and ΔPXX UBQLN2 for binding to Omp25 (Appendix Fig. S5E). We conclude that the SH3 domains serve as an inert barcode for our binding assay.

## Ubiquilin paralogs show substantial overlap in substrate binding specificity

To test the substrate binding preference of ubiquilins, we constructed a total of four substrate series, two based on the mitochondrial tail-anchored proteins Omp25 (Fig. 3D–F) and Tom5 (Appendix Fig. S6A), and two based on the ER tail-anchored proteins Sec61β (Appendix Fig. S7A) and VAMP2 (Appendix Fig. S8A). We chose tail-anchored proteins as our model substrates because they are verified ubiquilin substrates, and the single TMD simplifies data analysis. The wild-type TMD sequence was placed in

the middle of the series with a barcode of either 2 or 3, and the hydrophobicity of the TMD within a substrate series was varied by mutating residues to Leucine or Alanine such that the GRAVY score shifted by 0.2-0.3 for each substrate. The wild-type sequences of the four series have a broad range of hydrophobicity values, with GRAVY scores ranging from 1.61 to 3.29.

The Omp25 and Sec61β substrate series showed that ubiquilins bound preferentially to TMDs with moderate hydrophobicity (Fig. 3D–F; Appendix Fig. S7B,C). This trend was conserved between UBQLN1, 2, and 4. However, the TMD hydrophobicity that elicited maximum ubiquilin binding differed between the two substrate series. The Tom5 substrate series was the least hydrophobic substrate series. It had the lowest overall level of binding. It still showed a preference for moderately hydrophobic TMDs (Appendix Fig. S6B,C), but the trend was much less pronounced than with Omp25 and Sec61β. The VAMP2 substrate series was the most hydrophobic series and, unexpectedly, showed the highest overall level of substrate binding, with less hydrophobic sequences having the strongest binding (Appendix Fig. S8B,C). Overall, our results show that UBQLN1, 2, and 4 show a general trend for binding to TMDs with moderate hydrophobicity. However, the TMD GRAVY score that elicited maximum substrate binding varied based on the substrate series. This suggests that TMD GRAVY score alone fails to fully capture the biophysical parameters that govern substrate binding.

We observed modest, but reproducible differences in the substrate binding preferences of the ubiquilin paralogs. UBQLN4 consistently had the lowest levels of substrate binding, with the results being most pronounced in the Omp25 (Fig. 3E,F) and Tom5 (Appendix Fig. S6B,C) substrate series. Within the VAMP2 substrate series, which was the most hydrophobic series, there were no statistically significant differences in substrate binding (Appendix Fig. S8B,C).

UBQLN1 and UBQLN2 showed roughly equal levels of substrate binding, with the exception of the least hydrophobic substrate series, Tom5. However, as TMD hydrophobicity reached the more extreme values, UBQLN1 binding dropped close to background levels, whereas UBQLN2 binding remained significantly above background levels. This trend is clearest in the Omp25 substrate series (Fig. 3F), but is also observed with Sec61β (Appendix Fig. S7C) and Tom5 (Appendix Fig. S6C). Overall, we conclude that there is substantial redundancy in the substrate binding preferences of UBQLN1, 2, and 4, but UBQLN2 has the most robust substrate binding and the broadest substrate range, whereas UBQLN4 shows the lowest levels of substrate binding.

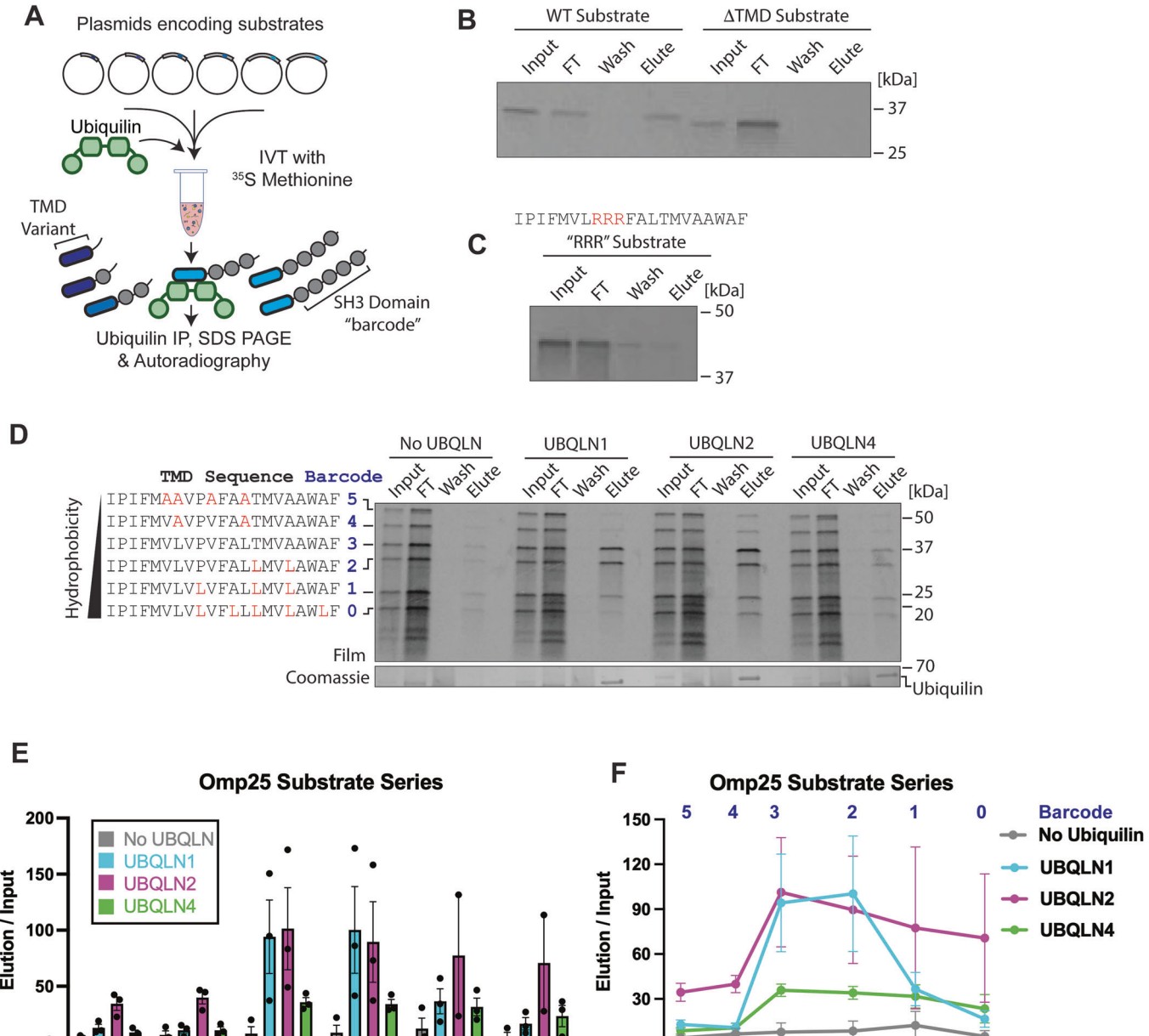

**Figure 3. Development of a barcoded binding assay to study ubiquilin-substrate binding.**

(A) Overview of the barcoded binding assay. A pool of six plasmids is simultaneously expressed by in vitro translation in the presence of 3 µM ubiquilin. Each plasmid contains a different TMD sequence and a variable number of SH3 domains to serve as a barcode. Substrate binding is assayed by ubiquilin immunoprecipitation. (B) There is no binding to the barcode 3 construct when the TMD is deleted. (C) Mutation of the Omp25 TMD with three consecutive Arginine residues is sufficient to disrupt binding to ubiquilin. (D) Barcoded binding assay with the Omp25 substrate series. The wild-type Omp25 TMD has a barcode of 3. Residues were mutated to either Leucine or Alanine to change TMD hydrophobicity. Note that the elution fraction is 5× more concentrated than the input fraction. (E) Quantification of (D), organized by barcode. Error bars are the standard error of the mean. A value of 100 corresponds to equal intensity of input and elution bands. (F) Quantification of (D), organized by TMD hydrophobicity (GRAVY). A higher GRAVY score indicates a more hydrophobic TMD. Barcode is noted at the top of the graph. Error bars are the standard error of the mean. Source data are available online for this figure.

## Ubiquilin STI1 domains selectively bind hydrophobic substrates with low helical propensity

Although ubiquilin binding within each substrate series showed maximal binding to TMDs with moderate hydrophobicity, the GRAVY score that elicited maximal binding differed between each substrate series. When all four substrate series are analyzed together, there is no discernible preference for moderately hydrophobic TMDs (Fig. 4A). This indicates that TMD GRAVY score alone is a poor predictor of ubiquilin binding.

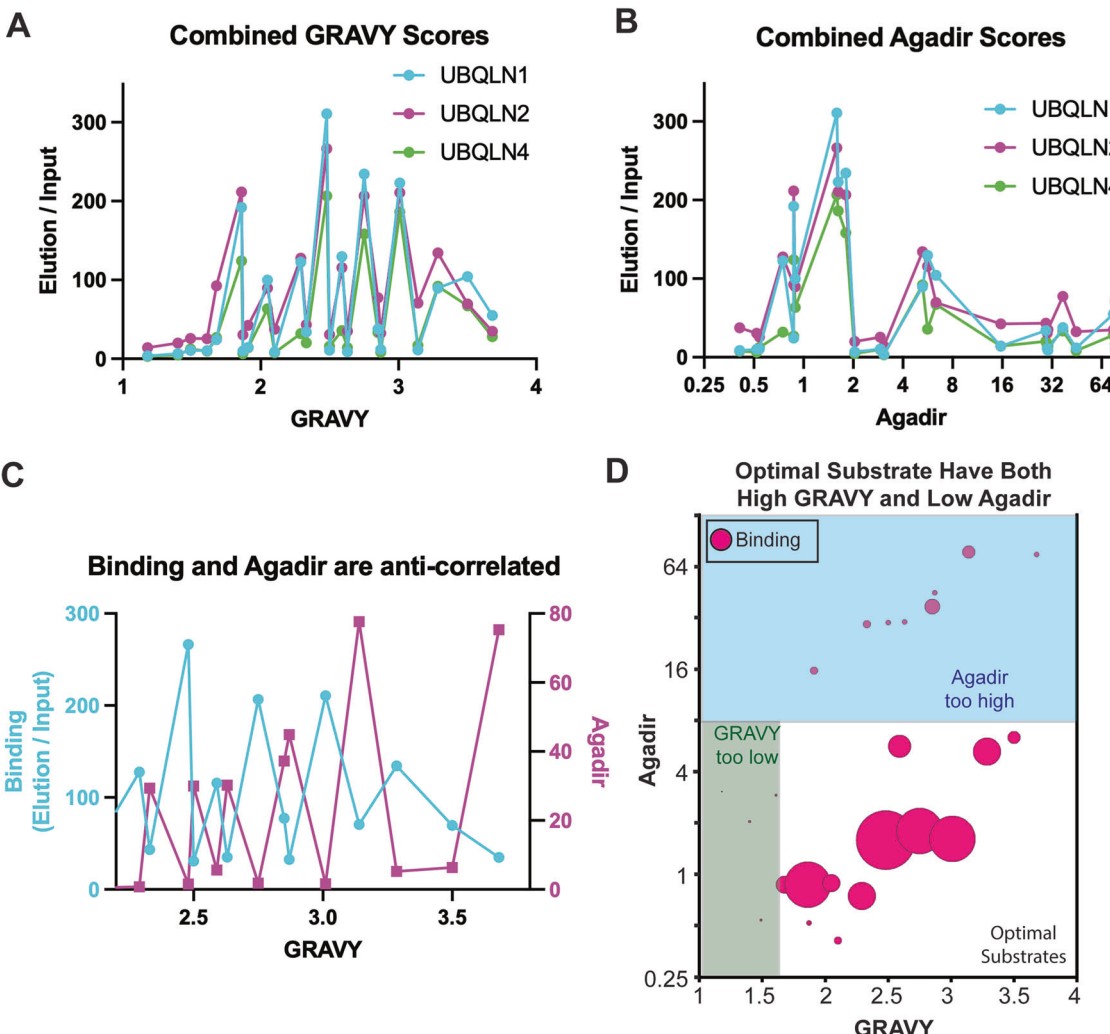

**Figure 4. Barcoded binding assay shows that ubiquilins bind to hydrophobic, disordered regions.**

(A) Combined data from all four substrate series does not show a clear pattern between TMD GRAVY score and ubiquilin binding. Each spot represents a different TMD construct from the barcoded binding assays with the Omp25, Sec61β, Tom5, and VAMP2 substrate series. (B) Combined data from all four substrate series does not show a clear pattern between the TMD Agadir score and ubiquilin binding. (C) Combined data from all four substrate series show that there is a strong anti-correlation between substrate binding to UBQLN2 and Agadir scores. (D) The bubble chart shows that optimal UBQLN2 substrates have both a high GRAVY score and a low Agadir score. Each spot represents a different TMD construct from the barcoded binding assays with the Omp25, Sec61β, Tom5, and VAMP2 substrate series. The radius of each spot corresponds to the amount of substrate within the elution fraction, with larger spots indicating better binding to UBQLN2. Source data are available online for this figure.

Previous studies had demonstrated that ER TA proteins have higher TMD hydrophobicity and higher helical propensity than mitochondrial TA proteins(Rao et al, 2016). As ubiquilins were shown to preferentially bind to mitochondrial membrane proteins (Whiteley et al, 2017; Itakura et al, 2016), we asked if helical propensity also governed ubiquilin substrate binding. The helical propensity of each TMD was calculated using the Agadir score, which uses helix/coil transition theory to provide an energetic description of a peptide's tendency to fold into an α-helix (Muñoz and Serrano, 1994, 1995). A high Agadir score corresponds to a high helical propensity. There was no obvious correlation between Agadir score and ubiquilin binding (Fig. 4B).

We next asked if a combination of TMD GRAVY and Agadir scores is a better predictor of ubiquilin binding. When plotted together, we observed a striking anti-correlation between substrate

binding to ubiquilins and Agadir score (Fig. 4C). When plotted as a bubble chart, we observed that optimal ubiquilin substrates are hydrophobic TMDs with a low helical propensity (Fig. 4D). As many hydrophobic residues also have high helical propensity, this could account for the previous observation that ubiquilins preferentially bind to moderately hydrophobic TMDs. We conclude that ubiquilins have no inherent defect in binding to highly hydrophobic TMDs, and substrate binding requires a combination of sufficient hydrophobicity and low helical propensity.

## Internal ubiquilin sequences bind to the STI1 hydrophobic groove

Having identified biophysical features that define optimal ubiquilin substrates, we next asked what prevents the thermodynamically

unfavorable solvent exposure of the STI1 hydrophobic groove in the absence of substrate. Acharya et al recently identified several sequences within *S. cerevisiae* Dsk2 that interact with the STI1 domain (Acharya et al, 2026). Interestingly, these placeholder sequences are in the intrinsically disordered regions of Dsk2. Given our results that ubiquilin STI1 domains preferentially bind to regions of low helicity and the observation that ubiquilins have several long stretches of disordered sequence, we hypothesized that ubiquilins contain similar internal sequences capable of self-interaction with the STI1 domain.

To test this hypothesis, we first asked if the placeholder sequences identified in *S. cerevisiae* Dsk2 are conserved in *H. sapiens* UBQLN2. Despite the significant size difference between UBQLN2 (624 residues) and Dsk2 (373 residues), we observed moderate conservation of these sequences (Appendix Fig. S9). We will refer to these putative placeholder sequences as PH1, 2, and 3. An analysis of the UBQLN2 AlphaFold-predicted structures shows that putative placeholders PH1 and PH2 have a modest helical content and border unstructured regions (Fig. 5A).

We next examined the biophysical properties of PH1, 2, and 3 and observed that the putative placeholder sequences all have low Agadir scores. Our model predicts that placeholder sequences must have suboptimal binding characteristics, otherwise the high effective concentration of the placeholder sequence will prevent exogenous sequences from binding to the STI1 domain. Consistent with our model, PH1 and PH2 are generally hydrophobic, but the GRAVY scores fall outside the optimal range for a true substrate (Fig. 5B). By contrast, the overall sequence of PH3 is not hydrophobic and has a negative GRAVY score. Closer examination of the AlphaFold model reveals that PH3 forms a short, broken amphipathic helix with the hydrophobic face perfectly positioned to interact with the hydrophobic groove of STI1-II (Appendix Fig. S10).

To test if PH1, 2, and 3 can bind to the STI1 domain, we again used the PURExpress IVT system to synthesize the placeholder sequences in the presence of physiological concentrations of UBQLN2. We observed robust binding of all 3 placeholder sequences to WT UBQLN2 and minimal binding in the no ubiquilin control (Fig. 5C).

To test how ALS mutations in the STI1 domains affect placeholder binding, we repeated the assay with the P189T, P440L, and M446R UBQLN2 mutants. Unlike the results with the hydrophobic TMD Omp25, we observed that the P189T and P440L mutations each led to a partial loss of binding to PH1, suggesting that PH1 binds to both the STI1-I and STI1-II domains with similar affinity (Fig. 5D). Conversely, the PH2 and PH3 sequences only showed disruption with the P189T mutation (Fig. 5E,F). We conclude that the interaction of STI1 domains with internal placeholder sequences is conserved from yeast to humans, that both STI1 domains in UBQLN2 can interact with internal placeholder sequences, and that ALS-causing mutations within the STI1 domains disrupt interaction with placeholder sequences.

## The PXX repeat region binds to STI1 domains

Given our results that ALS mutations in the STI1 domain disrupt the STI1-placeholder interaction, we hypothesized that the UBQLN2 PXX repeat region, an ALS mutation hotspot, also interacts with one of the STI1 domains. Examination of the Agadir and GRAVY scores shows biophysical parameters consistent with the other placeholder sequences identified above (Fig. 6A). To test this hypothesis, we again used our PURExpress binding assay. Consistent with our hypothesis, we observed robust binding of the PXX repeat region to WT UBQLN2 and minimal binding to the no ubiquilin control (Fig. 6B). We then repeated the assay with the P189T, P440L, and M446R UBQLN2 mutants (Fig. 6C). We observed a strong decrease in binding with the P189T mutation. We conclude that the PXX repeat region can bind to the STI1-I domain.

# Discussion

The STI1 domains have been linked to numerous roles in ubiquilin function and disease, such as chaperone activity, dimerization, phase separation, and ALS (Zheng et al, 2020; Safren et al, 2024; Itakura et al, 2016). However, a clear mechanistic and structural explanation for these multifaceted roles remains elusive. Here, we address these key questions by solving the first crystal structure of a ubiquilin family STI1 domain bound to a TMD and developing a novel barcoded binding assay to directly measure substrate interaction with the STI1 domains. Our structure shows that the STI1 domain binds to substrates via a methionine-rich hydrophobic groove. The barcoded binding assay demonstrates that STI1 domains preferentially bind to hydrophobic sequences with low helicity, two motifs present throughout the long, disordered regions between annotated domains in ubiquilins. Consistent with these results, we show that multiple conserved internal placeholder sequences, including the ALS-associated PXX repeat region, bind to the STI1 domains. Finally, we demonstrate that multiple ALS-causing mutations interfere with the STI1 hydrophobic groove, thereby disrupting binding to both TMDs and placeholder sequences. Together, these results lead to a new paradigm for understanding the role of the STI1 domains in ubiquilin phase separation and how this activity is dysregulated in ALS.

Our crystal structure shows a methionine-rich hydrophobic groove, which readily explains ubiquilin chaperone activity. In the absence of a bound TMD, the hydrophobic groove within each STI1 domain will have thermodynamically unfavorable interactions with solvent. Many TMD chaperones therefore use internal placeholder sequences to occupy the binding sites and act as a substrate selectivity filter that is displaced upon encounter with a bona fide substrate (Mateja et al, 2015; Rao et al, 2016; Voorhees and Hegde, 2016). Acharya et al recently identified several sequences within *S. cerevisiae* ubiquilin homolog Dsk2 that bind to the STI1 domain (Acharya et al, 2026). Here, we demonstrate that similar placeholder sequences also exist in UBQLN2, and these sequences bind to both STI1-I and STI1-II domains (Fig. 7A,B). Our results suggest that the placeholder-STI1 interaction is conserved from yeast to humans and is perhaps a general feature of STI1 domain-containing proteins.

The placeholder-STI1 interaction has profound implications for ubiquilin activity, particularly with substrate binding, ubiquilin phase separation, and ALS. Based on the results of the barcoded binding assay, the biophysical properties of the placeholder sequences appear to be suboptimal for robust STI1 domain interaction. However, the presence of multiple internal placeholder

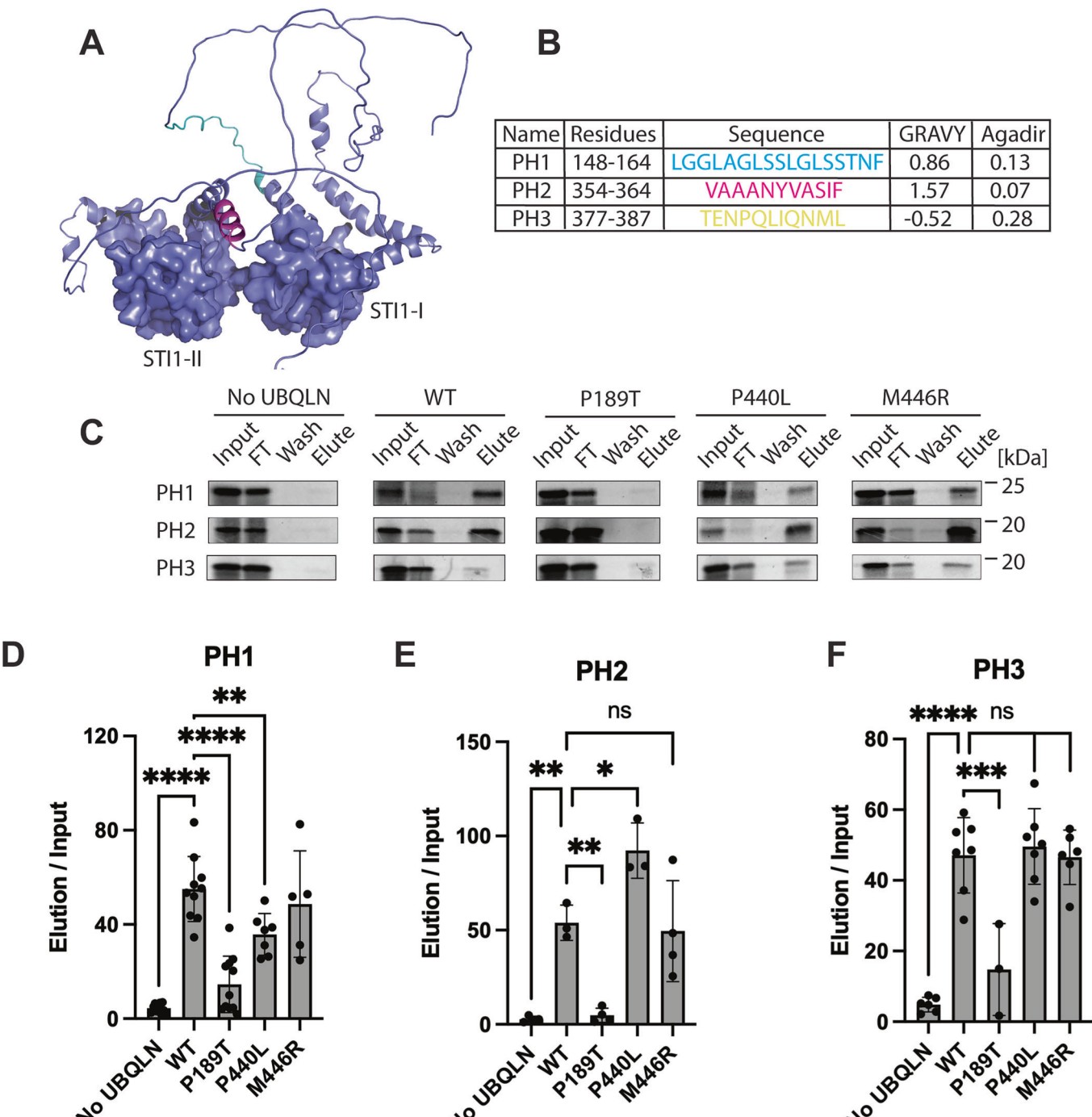

**Figure 5. Internal placeholder sequences bind to the STI1 hydrophobic groove.**

(A) AlphaFold model of UBQLN2 shows that placeholder sequences 1 and 2 have modest helical content and are adjacent to unstructured regions. PH1 and PH2 are colored cyan and magenta, respectively. STI1 domains are shown as surfaces. UBL and UBA domains are omitted for clarity. (B) Summary of biophysical properties for UBQLN2 placeholder sequences. (C) Representative results of placeholder 1, 2, or 3 binding to UBQLN2 variants. A binding assay was performed with 3 μM UBQLN2. (D) Quantification of (C), $N \geq 5$. Error bars are standard deviation centered on the mean. $P$ values were calculated by one-way ANOVA with Dunnett's post hoc test; ****$P < 0.0001$, ***$P < 0.001$, **$P < 0.01$, *$P < 0.05$, ns: not significant. Exact $P$ values: No UBQLN $= 3.6 \times 10^{-11}$, P189T $= 1.1 \times 10^{-8}$, P440L $= 0.0093$, and M446R $= 0.75$. (E) Quantification of (C), $N \geq 3$. Error bars are standard deviation centered on the mean. $P$ values were calculated by one-way ANOVA with Dunnett's post hoc test; **$P < 0.01$, *$P < 0.05$, ns: not significant. Exact $P$ values: No UBQLN $= 0.0018$, P189T $= 0.0025$, P440L $= 0.023$, and M446R $= 0.98$. (F) Quantification of (C), $N \geq 3$. Error bars are standard deviation centered on the mean. $P$ values were calculated by one-way ANOVA with Dunnett's post hoc test; ****$P < 0.0001$, ***$P < 0.001$, ns: not significant. Exact $P$ values: No UBQLN $= 6.9 \times 10^{-8}$, P189T $= 0.00012$, P440L $= 0.97$, and M446R $= 0.99$. Source data are available online for this figure.

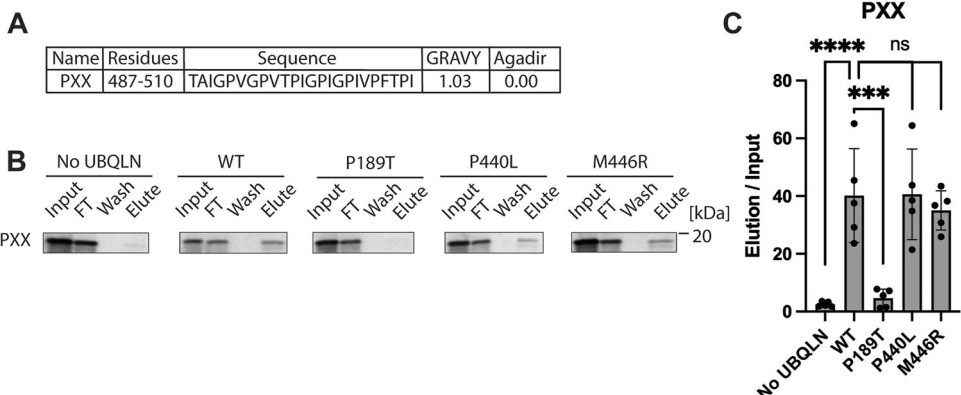

**Figure 6. The PXX repeat region predominantly binds to the STI1-I domain.**

(A) The PXX repeat region is hydrophobic (high GRAVY score) and has low helicity (low Agadir score), consistent with other placeholder sequences. (B) Representative results of the PXX repeat region binding to UBQLN2 variants. Binding assay was performed with 3 μM UBQLN2. (C) Quantification of (B), $N \geq 3$, error bars are the standard deviation centered on the mean. $P$ values were calculated by one-way ANOVA with Dunnett's post hoc test; ****$P < 0.0001$, ***$P < 0.001$, ns: not significant. Exact $P$ values: No UBQLN $= 6.7 \times 10^{-5}$, P189T $= 0.00014$, P440L $= 0.99$, and M446R $= 0.86$. Source data are available online for this figure.

sequences will lead to a high effective concentration, such that the STI1 domain is predominantly occupied by a placeholder sequence. A particularly enticing example is PH3, which is perfectly positioned as an amphipathic helix above STI1-II. This highly effective concentration may explain why substrates preferentially bind to the STI1-I domain. For an exogenous substrate, such as a TMD, to stably bind to the STI1 domain, it must have biophysical properties that allow it to outcompete the high effective concentration of the placeholder sequences. In this way, the placeholder sequences may serve as a selectivity filter to regulate substrate binding to the STI1 domain. Another important implication of our model is that substrate binding could be inhibited either by disrupting the hydrophobic groove or by stabilizing the placeholder-STI1 interaction. Differentiating between these models will be an important area of further investigation.

A key driver of phase separation is multivalency. It is already well documented that the ubiquilin UBL and UBA domains are important for self-interaction; however, it is less clear why the STI1-II domain is essential for phase separation. We propose that the STI1 domain contributes to phase separation in multiple ways. First, by the interaction of the STI1 domain with multiple internal placeholder sequences, and second, by STI1 dimerization (Fig. 7C). The STI1-placeholder interactions can occur intra- or intermolecularly, with intermolecular interactions becoming more prevalent as the total ubiquilin concentration increases. The displacement of internal placeholder sequences by a bona fide substrate implies that substrate binding will modulate ubiquilin phase separation by disrupting the STI1-placeholder interaction. Directly testing this model will be important for solidifying our understanding of ubiquilin phase separation.

There are likely more placeholder sequences in ubiquilins than those identified here, and it is possible that such placeholder sequences interact with additional ubiquilin domains. Indeed, paramagnetic spin-labeling studies have shown that the UBQLN2 UBAA motif interacts with the STI1-II domain (Dao et al, 2019). Differences in the number and/or biophysical properties of the placeholder sequences could also explain the different phase

separation properties of UBQLN1, UBQLN2, and UBQLN4 despite high sequence similarity in the STI1 domains. In particular, our data showing that the PXX repeat region serves as a putative placeholder could explain why UBQLN2 has a higher phase separation propensity than other ubiquilin paralogs (Dao et al, 2024). Identification of all placeholder sequences will be important for fully modeling ubiquilin phase separation.

The dimerization of the STI1 domains also contributes to the multivalency of human ubiquilins. Indeed, previous work has shown that the STI1-II domain is required for UBQLN2 dimerization (Dao et al, 2024). Furthermore, Acharya et al demonstrated that deletion of the STI1 domain from the *S. cerevisiae* Dsk2 has a larger effect on phase separation than co-deletion of three placeholder sequences (Acharya et al, 2026). The multiple contributions of the STI1 domain to intermolecular interactions explain why the STI1-II domain, but not the placeholder PXX region, is critical for ubiquilin phase separation (Dao et al, 2018). While our crystal structure of the *M. bicuspidata* Dsk2 STI1 domain excitingly shows a dimer in the asymmetric unit, care needs to be taken in extrapolating these results to human ubiquilins. Recent work has shown that, contrary to human ubiquilins, *S. cerevisiae* Dsk2 does not form stable dimers, although there is evidence that the STI1 domain is critical for self-interaction (Acharya et al, 2026). Consistent with these observations, mutational analysis of our crystallization construct shows that the TMD is a major driver of oligomerization. More work is needed for a full structural understanding of the STI1 dimerization in human ubiquilins.

The oligomeric state of the STI1 domain may also influence the biophysical properties of the optimal ubiquilin substrate. For example, a single STI1 domain may preferentially bind to an amphipathic helix, with the hydrophobic face buried in the hydrophobic groove and the hydrophilic face solvent-exposed. The presence of an amphipathic TMD is another differentiating feature between mitochondrial and ER TA proteins (Fry et al, 2021b), and may contribute to the apparent selectivity of ubiquilins for mitochondrial membrane proteins. Systematically testing if substrate binding to the STI1 domain requires a fully hydrophobic

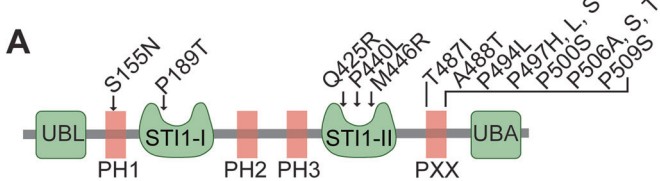

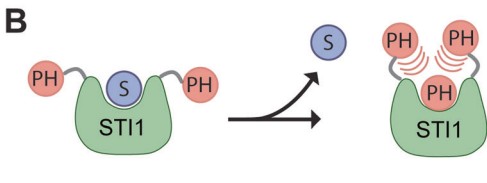

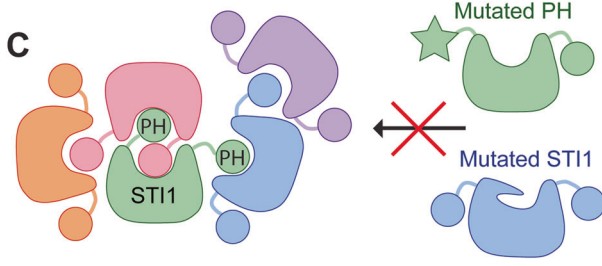

**Figure 7. Proposed model for how ALS mutations lead to altered phase separation.**

(A) Domain diagram of UBQLN2. The four identified placeholder sequences capable of interacting with the STI1 domains are shown in red. ALS-causing mutations within the placeholder sequences and STI1 domains are indicated. (B) Hydrophobic substrates (S) bind to the STI1 domain. In the absence of substrate, placeholders dynamically bind to the STI1 domain to limit solvent exposure of the hydrophobic groove. (C) Model for how intermolecular interactions between STI1 domains and placeholders contribute to the multivalency required for phase separation. ALS mutations within the placeholder sequences or STI1 domains are predicted to modulate this interaction, leading to altered phase separation. For visual clarity, each molecule of ubiquilin is a different color and is represented with only one STI1 domain and two placeholders.

TMD or an amphipathic helix will provide further insights into the mechanism of substrate selectivity.

Many UBQLN2 ALS mutations lead to altered phase separation properties, but the mechanistic details of how this occurs have remained obscure. Our model now makes testable, structure-based predictions for how ALS mutations modulate ubiquilin phase separation. Mutations in the STI1 domain are predicted to alter the hydrophobic groove. Consistent with these predictions, we demonstrate that the pathogenic P189T and P440L mutations disrupt placeholder binding to the STI1 domains. The interaction of the PXX region, with the STI1 domains is particularly exciting as numerous ALS mutations are contained within this sequence (Fig. 7A). Modulation of the STI1-placeholder interaction therefore presents a new paradigm for therapeutic intervention in ALS patients with UBQLN2 mutations.

In summary, our findings provide a structural and mechanistic understanding of the STI1 domains in ubiquilin proteins, elucidating their roles in substrate binding, phase separation, and ALS pathogenesis. By revealing the conserved interplay between STI1 domains and placeholder sequences, as well as the effects of ALS-

related mutations, this study highlights the critical role of hydrophobic interactions in regulating ubiquilin function. These insights lay the groundwork for future investigations into ubiquilin-mediated cellular processes and potential therapeutic interventions for ALS and related disorders

# Methods

**Reagents and tools table**

| Reagent/resource | Reference or source | Identifier or catalog number |
|---|---|---|
| **Experimental models** | | |
| T-Rex-293 Cell line | Invitrogen | Cat. #R71007 |
| BL21 CodonPlus (DE3) pRIL | Agilent | Cat. #230245 |
| **Recombinant DNA** | | |
| See Table 2 for a complete list of plasmids used in this study | | |
| **Antibodies** | | |
| Mouse monoclonal anti-Flag | GenScript | Cat. #A00187; RRID:AB_1720813 |
| Mouse monoclonal anti-actin | Invitrogen (viaThermoFisher) | Cat. #MA1-744; RRID:AB_2223496 |
| Mouse monoclonal anti-HA | Invitrogen (viaThermoFisher) | Cat. #26183; RRID:AB_10978021 |
| Goat polyclonal HRP anti-mouse | Invitrogen (viaThermoFisher) | Cat. #31430; RRID:AB_228307 |
| Goat polyclonal HRP anti-rabbit | Proteintech | Cat. #SA00001-2; RRID:AB_2722564 |
| **Chemicals, enzymes, and other reagents** | | |
| Super Signal West Femto Maximum Sensitivity Substrate | Thermo Scientific | Cat. #34095 |
| 4–20% Criterion TGX Stain-Free Protein Gel | Bio-Rad | Cat. #5678095 |
| Anti-DYKDDDDK Magnetic Agarose | Pierce (via ThermoFisher) | Cat. #PIA36797 |
| 3X (DYKDDDDK) Peptide | APExBIO | Cat. #A6001 |
| Blasticidin | Gibco (via ThermoFisher) | Cat. #A1113902 |
| Pen/Strep | Gibco (via ThermoFisher) | Cat. #15140122 |
| DMEM | Gibco (via ThermoFisher) | Cat. #10569044 |
| OptiMEM | Gibco (via ThermoFisher) | Cat. #51985034 |
| FBS | Phoenix Scientific | Cat. #PS-100-02-500 |
| Lipofectamine 2000 | Invitrogen (viaThermoFisher) | Cat. #11668019 |
| Universal nuclease | Pierce | Cat. #88702 |
| Halt™ Protease Inhibitor Cocktail (100x) | Thermo Scientific | Cat. #78430 |
| Tris-HCl | VWR | Cat. # VWRB85827-1KG |
| Bovine Serum Albumin | Fisher Scientific | Cat. # BP1605-100 |

| Reagent/resource | Reference or source | Identifier or catalog number |
|---|---|---|
| Sodium Dodecyl Sulfate | Thermo Fisher | Cat. #BP8200500 |
| Sodium Chloride | Fisher | Cat. # S271-10 |
| Coomassie Blue | Fisher Scientific | Cat. # BP101-50 |
| Glacial Acetic Acid | Spectrum Chemical MFG Corp | Cat. # A10102.5LTGL |
| Methanol | Fisher Scientific | Cat. # A411-4 |
| Hepes | Fisher | Cat. # BP310-5 |
| Sodium Phosphate Monobasic | VWR | Cat. # BDH9298-2.5KG |
| Sodium Phosphate Dibasic | VWR | Cat. # VWRV0404-2.5KG |
| Terrific Broth | Fisher | Cat. # BP246810 |
| X-ray Film | Fisher Scientific | Cat. # 248300 |
| X-ray Film Developer | Fisher Scientific | Cat. # 4010D |
| X-ray Film Fixer | Fisher Scientific | Cat. # 4010 F |
| PurExpress In Vitro protein synthesis kit | NEB | Cat. # E6800L |
| isopropyl β-D-1-thiogalactopyranoside | RPI corp | Cat. # I56000-5.0 |
| Phenylmethylsulfonyl fluoride | RPI corp | Cat. # P20270-5.0 |
| Universal Cell Nuclease | Pierce | Cat. # 88700 |
| Lysozyme | Millipore Sigma | Cat. # L6876-25G |
| EasyTag L-[$^{35}$S] methionine stabilized aqueous solution | Perkin Elmer | Cat. # NEG709A005MC |
| **Software** | | |
| ImageJ | https://imagej.net/ij/ | |
| GraphPad Prism | https://www.graphpad.com/features | |
| GRAVY Score Calculation | https://www.gravy-calculator.de/ | |
| Agadir Score Calculation | http://agadir.crg.es/ | |
| **Other** | | |
| Imager | Bio-Rad | ChemiDoc Touch Imaging System |
| X-Ray Film Processor | Konica | SRX-101A |
| Gel Dryer | Bio-Rad | Model Gel Dryer 583 |
| FPLC system | Bio-Rad | NGC Quest 10 plus |
| SEC Column | Cytiva | Superdex 200 increase 10/300 GL |
| Amicon Ultra centrifugal filter | Millipore Sigma | UFC805008 |

## Plasmids

All plasmids are listed in Table 2. The barcoded substrates have the general structure of 3xHA-Sec61β soluble domain-(SH3)$_n$-TMD-3F4, where n varies between 0 and 5 to generate the barcode. The

Omp25 substrates were generated by gene synthesis (Azenta) and cloned into the PURExpress DHFR Control Vector (NEB) between the NdeI and NotI sites. Note that each SH3 domain had a unique DNA sequence and unique restriction enzyme sites between the SH3 domains and the TMD. Subsequent substrate series were generated by replacing the TMD sequence with a new TMD sequence generated by gene synthesis. New TMD sequences were restriction-cloned between KpnI and BamHI. Ubiquilin variants were cloned into the pET28a vector and have the general structure of His$_6$-3C-3xFLAG-ubiquilin (Gaur and Wohlever, 2025).

For binding assays with a single substrate, the sequence of interest (Omp25 TMD, PXX repeat region, or placeholder sequence) was cloned into the barcode 0 or 1 construct between the KpnI and BamHI sequences. The final construct had the general structure 3xHA-Sec61β soluble domain-test sequence-3F4. For cell culture experiments, 3xHA-UBQLN2 variants or 3xFLAG-Omp25 were cloned into the pUdOm1.0 vector (Manigat et al, 2024) under control of the CMV promoter.

The crystallization construct was cloned into a pET28b vector with an N-terminal His$_6$ tag followed by a 3 C protease site (Gaur and Wohlever, 2025). The fusion construct consisted of the STI1 domain of M. bicupsidata Dsk2 (residues 165–234) followed by the receiver domain of *M. xanthus* FrzS (residues 3-124) and a variant of the VAMP2 TMD (MMIALGVACAIALAIAAVYF). The VAMP2 TMD is the same sequence used in the barcode 5 data, which showed the strongest binding to ubiquilins (Appendix Fig. S8). The linker between the STI1 and FrzS domain is WGS, and the linker between the FrzS domain and TMD is GNS.

## Expression and purification of ubiquilins

Ubiquilins were expressed in BL21 DE3 cells with the pRIL plasmid (Agilent). Cells were grown in terrific broth at 37 °C until OD$_{600}$ = 0.6-0.8. Ubiquilin protein expression was induced by the addition of 0.5 mM isopropyl β-D-1-thiogalactopyranoside (IPTG, RPI Corp.). Upon induction, the temperature was dropped to 16 °C, and the cells were incubated for 16 h. Cells were pelleted by centrifuging for 20' at 3900 rpm in an Eppendorf 5810R centrifuge. The cell pellet from 1 L of culture was resuspended in 50 mL of UBQLN Lysis Buffer (50 mM sodium phosphate pH 7.5, 150 mM NaCl, 0.01 mM EDTA, 10% glycerol), supplemented with 0.05 mg/mL lysozyme (Sigma) and 1 mM Phenylmethylsulfonyl fluoride (PMSF). Samples were stored at −80 °C until purification.

Cell pellets were thawed in water and then kept on ice throughout the procedure to minimize proteolysis. After thawing, 2 μL of Universal Nuclease (Pierce) was added to each cell pellet, and the volume was brought to ~120 mL by addition of lysis buffer or additional cell pellets. Cells were transferred to a metal cup surrounded by ice and lysed by sonication. The supernatant was isolated by centrifugation at 18,500×g for 30' at 4 °C and purified by Ni-NTA affinity chromatography (Thermo Fisher) on a gravity column in the cold room. Ni-NTA resin was washed with 25 column volumes (CV) of UBQLN Lysis Buffer. The sample was eluted with Lysis Buffer supplemented with 250 mM imidazole. The eluent was concentrated to < 1 mL in a 50 kDa MWCO Amicon Ultra centrifugal filter (Millipore). The sample was then incubated overnight at 4 °C with a 100:1 ratio of ubiquilin to 3 C protease to remove the His$_6$ tag.

The protein was further purified by size-exclusion chromatography (SEC) on a Superdex 200 Increase 10/300 GL, GE

**Table 2.  Plasmids used in this study.**

| Plasmid | Description | Reference |
|---|---|---|
| pJO_79 | 3xHA-Sec61β soluble domain-$(SH3)_0$-Omp25TMDVariant1-3F4 | This study |
| pJO_80 | 3xHA-Sec61β soluble domain-$(SH3)_1$-Omp25TMDVariant2-3F4 | This study |
| pJO_81 | 3xHA-Sec61β soluble domain-$(SH3)_2$-Omp25TMDVariant3-3F4 | This study |
| pJO_82 | 3xHA-Sec61β soluble domain-$(SH3)_3$-Omp25TMDVariant4-3F4 | This study |
| pJO_83 | 3xHA-Sec61β soluble domain-$(SH3)_4$-Omp25TMDVariant5-3F4 | This study |
| pJO_84 | 3xHA-Sec61β soluble domain-$(SH3)_5$-Omp25TMDVariant6-3F4 | This study |
| pJO_120 | 3xHA-Sec61β soluble domain-$(SH3)_0$-Sec61βTMDVariant1-3F4 | This study |
| pJO_121 | 3xHA-Sec61β soluble domain-$(SH3)_1$-Sec61βTMDVariant2-3F4 | This study |
| pJO_122 | 3xHA-Sec61β soluble domain-$(SH3)_2$-Sec61βTMDVariant3-3F4 | This study |
| pJO_123 | 3xHA-Sec61β soluble domain-$(SH3)_3$-Sec61βTMDVariant4-3F4 | This study |
| pJO_124 | 3xHA-Sec61β soluble domain-$(SH3)_4$-Sec61βTMDVariant5-3F4 | This study |
| pJO_125 | 3xHA-Sec61β soluble domain-$(SH3)_5$-Sec61βTMDVariant6-3F4 | This study |
| pJO_126 | 3xHA-Sec61β soluble domain-$(SH3)_0$-Tom5TMDVariant1-3F4 | This study |
| pJO_127 | 3xHA-Sec61β soluble domain-$(SH3)_1$-Tom5TMDVariant2-3F4 | This study |
| pJO_128 | 3xHA-Sec61β soluble domain-$(SH3)_2$-Tom5TMDVariant3-3F4 | This study |
| pJO_129 | 3xHA-Sec61β soluble domain-$(SH3)_3$-Tom5TMDVariant4-3F4 | This study |
| pJO_130 | 3xHA-Sec61β soluble domain-$(SH3)_4$-Tom5TMDVariant5-3F4 | This study |
| pJO_131 | 3xHA-Sec61β soluble domain-$(SH3)_5$-Tom5TMDVariant6-3F4 | This study |
| pJO_132 | 3xHA-Sec61β soluble domain-$(SH3)_0$-VAMP2TMDVariant1-3F4 | This study |
| pJO_133 | 3xHA-Sec61β soluble domain-$(SH3)_1$-VAMP2TMDVariant2-3F4 | This study |
| pJO_134 | 3xHA-Sec61β soluble domain-$(SH3)_2$-VAMP2TMDVariant3-3F4 | This study |
| pJO_135 | 3xHA-Sec61β soluble domain-$(SH3)_3$-VAMP2TMDVariant4-3F4 | This study |
| pJO_136 | 3xHA-Sec61β soluble domain-$(SH3)_4$-VAMP2TMDVariant5-3F4 | This study |
| pJO_137 | 3xHA-Sec61β soluble domain-$(SH3)_5$-VAMP2TMDVariant6-3F4 | This study |
| pJO_065 | 3xHA-Sec61β soluble domain-$(SH3)_3$-ΔTMD-3F4 | This study |
| pJO_161 | 3xHA-Sec61β soluble domain-$(SH3)_4$-Omp25TMD(RRR)-3F4 | This study |
| SP051 | 6xHis-3C-STI1-Frzs-TMD | This study |
| SP079 | 6xHis-3C-STI1-Frzs-ΔTMD | This study |
| SP086 | 6xHis-3C-STI1-Frzs-ΔTMD M29D, N37A, M71D | This study |
| SP049 | 6xHis-3C-3xFLAG-UBQLN2 | This study |
| SP036 | 6xHis-3C-3xFLAG-UBQLN2 P189T | This study |
| SP037 | 6xHis-3C-3xFLAG-UBQLN2 P440L | This study |
| SP029 | 6xHis-3C-3xFLAG-UBQLN2 M446R | This study |
| SP078 | 6xHis-3C-3xFLAG-UBQLN2 ΔPXX | This study |
| SP080 | 3xHA-Sec61β soluble domain-$(SH3)_0$-Omp25TMD(WT)-3F4 | This study |
| SP081 | 3xHA-Sec61β soluble domain-$(SH3)_1$-Omp25TMD(WT)-3F4 | This study |
| SP082 | 3xHA-Sec61β soluble domain-$(SH3)_2$-Omp25TMD(WT)-3F4 | This study |
| SP083 | 3xHA-Sec61β soluble domain-$(SH3)_3$-Omp25TMD(WT)-3F4 | This study |
| SP084 | 3xHA-Sec61β soluble domain-$(SH3)_4$-Omp25TMD(WT)-3F4 | This study |
| SP085 | 3xHA-Sec61β soluble domain-$(SH3)_5$-Omp25TMD(WT)-3F4 | This study |
| SP034 | 3xHA-Sec61β soluble domain-$(SH3)_1$-PH1-3F4 | This study |
| SP033 | 3xHA-Sec61β soluble domain-$(SH3)_0$-PH2-3F4 | This study |
| SP058 | 3xHA-Sec61β soluble domain-$(SH3)_0$-PH3-3F4 | This study |
| SP059 | 3xHA-Sec61β soluble domain-$(SH3)_0$-PXX-3F4 | This study |

**Table 2.** (continued)

| Plasmid | Description | Reference |
|---------|-------------|-----------|
| pST66 | 3xHA-UBQLN2 WT (pUdOM1.0) | This study |
| pST67 | 3xHA-UBQLN2 P189T (pUdOM1.0) | This study |
| pST89 | 3xHA-UBQLN2 P189T /Q425R (pUdOM1.0) | This study |
| pST90 | 3xHA-UBQLN2 P189T/M446R (pUdOM1.0) | This study |
| pST91 | 3xHA-UBQLN2 P189T/P440L (pUdOM1.0) | This study |
| pST70 | 3xFLAG-Omp25 (pUdOM1.0) | This study |

Healthcare in UBQLN FPLC Buffer (20 mM Tris pH 7.5, 100 mM NaCl). Peak fractions were pooled, concentrated to 5–15 mg/mL in a 50 kDa MWCO Amicon Ultra centrifugal filter (Millipore), and aliquots were flash-frozen in liquid nitrogen and stored at −80 °C. Protein concentrations were determined by $A_{280}$ using a calculated extinction coefficient (Expasy).

## Expression and purification of crystallization construct

The crystallization construct was grown and expressed as described above for ubiquilins, with the following changes. The crystallization construct was expressed in BL21 DE3 pRIL cells and grown in terrific broth until $OD_{600}$ of 0.6–0.8, at which point expression was induced 0.5 mM IPTG at room temperature for 16 h. Lysis Buffer is 50 mM Tris pH 7.5, 20 mM Imidazole, 500 mM NaCl, 10% glycerol, 1 mM DTT, and Elution Buffer is 50 mM Tris pH 7.5, 500 mM imidazole, 150 mM NaCl, 1 mM DTT. After elution, the eluent was concentrated to <1 mL in a 10 kDa MWCO Amicon Ultra centrifugal filter (Millipore). The sample was then mixed with a 100:1 ratio of protein to 3C protease and dialyzed overnight at 4 °C in Dialysis Buffer (50 mM Tris pH 7.5, 150 mM NaCl, 1 mM β-mercaptoethanol). The protein was further purified by size-exclusion chromatography (SEC) on a Superdex 200 Increase 10/300 GL column equilibrated in Dialysis Buffer. Peak fractions were pooled, concentrated to >14 mg/mL in a 10 kDa MWCO Amicon Ultra centrifugal filter (Millipore), and aliquots were flash-frozen in liquid nitrogen and stored at −80 °C. Protein concentrations were determined by $A_{280}$ using a calculated extinction coefficient (Expasy).

## Protein crystallization and structure determination

Crystals of STI1-Frzs-TMD construct were grown at 4 °C using hanging-drop vapor diffusion method by mixing equal volume (1 μL) of protein and reservoir solution containing 2.5% MPD, 100 mM Tris pH 8.5. Large crystals were grown over a period of three days and shown to belong to the space group P12₁1. Protein crystals were then soaked 2–3 times in cryoprotectant containing the well solution and 20% glycerol and flash-frozen in liquid nitrogen.

Diffraction data were collected at Beamline 8.2.2 at Advanced Light Source at Lawrence Berkeley National Laboratory (Berkeley, CA). Data were processed and scaled using Mosflm (Battye et al, 2011) and CCP4 (Agirre et al, 2023). Crystals of STI1-Frzs-TMD diffracted to 1.98 Å resolution and belong to space group P12₁1 with unit cell dimensions $a$ = 67.41 Å, $b$ = 63.92 Å, and $c$ = 67.43 Å. Phases were determined by molecular replacement using Phaser-

MR (McCoy et al, 2007) program incorporated in the Phenix package (Adams et al, 2010). The receiver domain from Myxococcus xanthus social motility protein FrzS (PDB: 2GKG) was used as the search model, which represents 55% of the STI1-Frzs-TMD construct. The initial solution from molecular replacement generated an interpretable electron density map. Model building and refinement were carried out using COOT (Emsley et al, 2010). The resulting model was further improved by positional and anisotropic B-factor refinement in Phenix, and the model quality was monitored throughout all stages of the refinement process and validated using MolProbity (Williams et al, 2018). The atomic coordinates and structure factors are deposited in the PDB: 9CKX.

## Size-exclusion chromatography

Analytical size-exclusion chromatography was performed on a Bio-Rad NGC Quest 10 Plus with a Superdex 200 Increase 10/300 GL column (Cytiva). A total of 0.5 mL of sample at 2 mg/mL was loaded onto the column at a flow rate of 0.5 mL/min. Aldolase (158 kDa), Conalbumin (75 kDa), Ovalbumin (44 kDa), and Lysozyme (14 kDa) were used as molecular weight standards.

## Barcoded binding assay

In Vitro Translation (IVT) was performed using the PURExpress in vitro protein synthesis kit (NEB). Each IVT reaction contained 4 μL of PURE Solution A, 3 μL of PURE Solution B, 1 μL of EasyTag L-[³⁵S] methionine stabilized aqueous solution (Perkin Elmer), 0.2 μL of Superase-in RNase inhibitor (Invitrogen), 120 ng of plasmid DNA, and a final concentration of 3 μM ubiquilin. Nuclease-free water was added to bring the total reaction volume to 10 μL. The reaction was incubated at 37 °C for 2 h.

The barcoded binding assay was performed with a mixture of each of the six plasmids such that there was roughly equal expression of each construct. The ratio of each plasmid was determined empirically by running individual IVT reactions for each plasmid within a substrate series. All six reactions were run on the same SDS PAGE gel and analyzed by autoradiography and ImageJ.

After the 2 h incubation, the IVT reaction was diluted with 40 μL of Wash Buffer (20 mM Tris pH 7.5, 100 mM NaCl, 1 mg/mL BSA). A 5 μL sample was taken as the "INPUT" fraction. The remaining 45 μL were added to 2 μL of settled anti-FLAG magnetic beads (4 μL of 50% slurry) (Thermo Fisher) that had been equilibrated in Wash Buffer. The sample was allowed to bind to resin for 30' at 4 °C before 5 μL of the sample was taken for the

"FLOW THROUGH" fraction, and the remaining flow-through was discarded. Beads were then washed 4× with 1 mL of wash buffer at 4 °C for 10'. After the final wash, 25 μL of the sample was taken for the "WASH" fraction. To elute ubiquilins, 25 μL of Wash Buffer containing 1 mg/mL 3x FLAG Peptide (ApexBio) was added, and the sample was incubated in the Thermomixer at 37 °C for 30'. All 25 μL were taken for the "ELUTE" fraction.

The "INPUT", "FLOW THROUGH", "WASH", AND "ELUTE" fractions were brought to a final volume of 25 μL by adding MilliQ Water, and then 8.8 μL of 4× SDS PAGE Loading Buffer was added to the samples. Samples were then run on a 4–20% gradient Criterion gel (Bio-Rad), stained with Coomassie, and imaged.

Gel was then dried either by incubating in drying solution (40% methanol, 10% glycerol, 7.5% acetic acid) for 5' and then loading into a gel drying apparatus (Promega) and allowed to sit at room temperature for 12–16 h with a fan on the apparatus. Alternatively, the gel was placed on filter paper and dried with a gel drier (Bio-Rad). Dried gel was then exposed to X-ray film (RPI Corp) for 24 h and then developed. Band intensity was measured using ImageJ.

Binding assays with a single substrate were carried out as described above, except that the plasmid DNA for a single substrate was used. The final reaction still contained 120 ng of plasmid DNA.

### Cell culture and cross-linking IP

HEK T-Rex 23 cells were grown in DMEM media with high glucose, GlutaMAX, and pyruvate (Gibco 10569044), supplemented with 10% FBS, and 100 U/mL penicillin, 100 μg/mL streptomycin, and 15 μg/mL blasticidin. Cell lines were authenticated by STR profiling and tested for mycoplasma contamination. Approximately 24 h prior to transfection, cells were split into 10 cm culture dishes with media lacking antibiotics. Immediately prior to transfection, cells were washed with Opti-MEM (Gibco 51985034) and then incubated with 5 mL of Opti-MEM. Co-transfections utilized 36 μg Omp25 of DNA, 30 μg of UBQLN2 DNA in 1 mL of Opti-MEM, and 36 μL of Lipofectamine 2000 (Invitrogen 11668019) in 1 mL Opti-MEM. DNA and Lipofectamine were mixed together for 20 min at room temperature and then added to the cells. Approximately 4–5 h post-transfection, Opti-MEM media was exchanged for DMEM supplemented with 10% FBS but lacking antibiotics.

Approximately 24 h post-transfection, cross-linking IP was performed by adding 5 mL of PBS containing 1 mM of DSP (Pierce 22585). After a 30-min incubation at room temperature, the reaction was quenched for 15 min by adding Tris-HCL buffer pH 7.5 to a final concentration of 20 mM.

Cells were lysed by mixing with the lysis buffer (25 mM Tris-HCl, pH 7.4, 150 mM NaCl, 1 mM EDTA, 1% NP-40, and 5% Glycerol) supplemented with 125 Units/mL of universal nuclease (Pierce 88702) and 1× Halt™ Protease Inhibitor Cocktail (Thermo-Scientific 78430). Lysates were centrifuged at 14,000 rpm for 15 min at 4 °C, and the supernatant was normalized by A280 with cell lysate dilution buffer (25 mM Tris-HCl, pH 7.4, 150 mM NaCl, 0.5 mM EDTA).

Cell lysate was mixed with 50 μL of anti-FLAG magnetic resin (Pierce PIA36797) equilibrated in FLAG IP Buffer (25 mM HEPES pH 7.5, 250 mM KOAc, 0.1% Tween 20). After a 1-h incubation, the flow through was discarded, and the resin was washed 3× with 500 μL of FLAG IP Buffer, followed by a 1× wash with 500 μL High Salt FLAG IP Buffer (25 mM HEPES pH 7.5, 500 mM NaCl,

0.5 mM EDTA, 1 mM PMSF). The sample was eluted by adding 50 μL of 1 mg/mL 3xFLAG peptide (ApexBio A6001) and incubating at 37 °C for 30 min with gentle shaking.

### Western blots

SDS PAGE was performed in a 4–20% gradient Criterion gel (Bio-Rad). Gels were transferred to 0.2 μM PVDF transfer membrane using the Trans-Blot Turbo Transfer System (Bio-Rad) for 10 min at mixed molecular weight settings. Membranes were blocked in 5% milk, antibody incubations were in 1% milk, and washing utilized 1× TBST (20 mM Tris-Base, 200 mM NaCl, 0.1%Tween-20, pH 7.4), all using standard techniques. Primary antibodies include anti-HA (Invitrogen 26183, 1:5000 dilution), anti-FLAG (GenScript A00187, 1:5000 dilution), and anti-Actin (Invitrogen MA1-744, 1:10,000 dilution). Blots were developed using SuperSignal West Femto Maximum Sensitivity Substrate (Thermo Scientific).

## Data availability

The data that support the findings of this study are openly available in Mendeley Data at https://data.mendeley.com/datasets/cx5cczftby/1 (https://doi.org/10.17632/cx5cczftby.1) and at the Protein Data Bank https://www.rcsb.org/structure/9CKX (PDB: 9CKX).

The source data of this paper are collected in the following database record: biostudies:S-SCDT-10_1038-S44318-026-00745-9.

## Peer review information

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

## Acknowledgements

The authors wish to thank members of the Castañeda and Wohlever labs for helpful discussions and feedback on the project. This work was supported by NSF CAREER Award 2343131 (MLW), NIH grant R35 GM137904-01S2 (MLW). The ALS ENABLE beamlines are supported in part by the National Institutes of Health, National Institute of General Medical Sciences, grant P30 GM124169-01. The Advanced Light Source is a Department of Energy Office of Science User Facility under Contract No. DE-AC02-05CH11231.

## Author contributions

**Joan Onwunma**: Conceptualization; Resources; Data curation; Formal analysis; Validation; Investigation; Visualization; Methodology; Writing—original draft. **Saeed Binsabaan**: Conceptualization; Resources; Data curation; Formal analysis; Validation; Investigation; Visualization; Methodology; Writing—review and editing. **Shawn P Allen**: Conceptualization; Resources; Data curation; Formal analysis; Validation; Investigation; Visualization; Methodology; Writing—review and editing. **Sachini R Thanthirige**: Conceptualization; Resources; Data curation; Formal analysis; Investigation; Visualization; Methodology; Writing—review and editing. **Deepika Gaur**: Resources; Investigation; Methodology; Writing—review and editing. **Banumathi Sankaran**: Resources; Data curation; Investigation; Writing—review and editing. **Matthew L Wohlever**: Conceptualization; Resources; Data curation; Supervision; Funding acquisition; Visualization; Methodology; Writing—original draft; Project administration; Writing—review and editing.

Source data underlying figure panels in this paper may have individual authorship assigned. Where available, figure panel/source data authorship is listed in the following database record: biostudies:S-SCDT-10_1038-S44318-026-00745-9.

## Disclosure and competing interests statement

The authors declare no competing interests.

