## [Peer Review File · The EMBO Journal]

ALS mutations disrupt self-association between the ubiquilin STI1 hydrophobic groove and internal placeholder sequences

Joan Onwunma, Saeed Binsabaan, Shawn Allen, Sachini Thanthirige, Deepika Gaur, Banumathi Sankaran, and Matthew Wohlever

Corresponding author(s): Matthew Wohlever (wohlever@pitt.edu)

Review Timeline:

Submission Date:	27th May 25
Editorial Decision:	8th Jul 25
Revision Received:	14th Jan 26
Editorial Decision:	16th Feb 26
Revision Received:	17th Feb 26
Accepted:	26th Feb 26

Editor: Hartmut Vodermaier

Transaction Report:

Dr. Matthew L Wohlever
University of Pittsburgh
Cell Biology
Pittsburgh, Pennsylvania 15261

8th Jul 2025

Re: EMBOJ-2025-121480
ALS mutations disrupt self-association between the Ubiquilin Sti1 domain and placeholder sequences

Dear Dr. Wohlever,

Thank you for submitting your manuscript on ubiquilin Sti1 domain structure for our editorial consideration. We sent your work to three expert reviewers, who have now returned the reports copied below. The referees appreciate your structural and biophysical analyses, including the newly developed barcoded binding assay for Sti1 domain interactions. However, they also raise a varying number of specific issues and open questions, whose clarification would be important to further strengthen the study and to make it a compelling candidate for EMBO Journal publication.

Should you be able to satisfactorily address these various concern, we would be open to considering a revised version further for The EMBO Journal. In particular, I would strongly encourage you to follow referee 3's suggestions for extending particular validations also to the utilized human UBQLNs and human cell lines, as well as domain-restricted constructs. Since it is our policy to allow only a single round of major revision, I would invite you to send me a preliminary point-by-point response already during the early stages of your revision work, so that we could discuss your revision plans for The EMBO Journal, and whether a less completely revised manuscript might alternatively be suitable for one of our sister journals like EMBO Reports or Life Science Alliance. We would also be open to extension of the default three-months revision period if needed; our 'scooping protection' (meaning that competing work appearing elsewhere in the meantime will not affect our considerations of your study) would of course remain valid throughout the whole period.

Detailed information on preparing, formatting and uploading a revised manuscript can be found below and in our Guide to Authors, and adhering to them as closely as possible shall greatly facilitate editorial processing upon resubmission. Thank you again for the opportunity to consider this work for The EMBO Journal, and I look forward to hearing from you in due time.

With kind regards,

Hartmut Vodermaier

3) Revised manuscript text (including main tables, and figure legends for main and EV figures) has to be submitted as editable

text file (e.g., .docx format). We encourage highlighting of changes (e.g., via text color) for the referees' reference.

4) Each main and each Expanded View (EV) figure should be uploaded as individual production-quality files (preferably in .eps, .tif, .jpg formats). For suggestions on figure preparation/layout, please refer to our Figure Preparation Guidelines:

8) Please note that supplementary information at EMBO Press has been superseded by the 'Expanded View' for inclusion of additional figures, tables, movies or datasets; with up to five EV Figures being typeset and directly accessible in the HTML version of the article. For details and guidance, please refer to:

embopress.org/page/journal/14602075/authorguide#expandedview

9) To facilitate reproducibility and cross-laboratory adoption of methodologies, please structure the Materials & Methods section as outlined in our guide to authors, including a completed Reagents and Tools Table that can be downloaded from our author guidelines as well (<https://www.embopress.org/page/journal/14602075/authorguide#structuredmethods>).

10) Digital image enhancement is acceptable practice, as long as it accurately represents the original data and conforms to community standards. If a figure has been subjected to significant electronic manipulation, this must be clearly noted in the figure legend and/or the 'Materials and Methods' section. The editors reserve the right to request original versions of figures and the original images that were used to assemble the figure. Finally, we generally encourage uploading of numerical as well as gel/blot image source data; for details see: embopress.org/page/journal/14602075/authorguide#sourcedata

Further information is available in our Guide For Authors:

Revision to The EMBO Journal should be submitted online within 90 days, unless an extension has been requested and approved by the editor; please click on the link below to submit the revision online before 6th Oct 2025:

Link Not Available

If you choose to alternatively have this study further considered by another EMBO Press publication, please use the following hyperlink to directly transfer the manuscript, optionally with inclusion of referee reports and identities:

Link Not Available

Referee #1:

Onwunma et al. is an elegant and thought-provoking biochemical investigation into UBQLN function through a very thorough characterization of the interactions of the Sti1 domain. The authors obtained a crystal structure of the yeast Dsk2 Sti1 domain bound to a transmembrane domain (TMD) at 1.98 Å, showing dimerization that creates a hydrophobic groove for TMD binding. A biochemical assay to probe the characteristics of TMD binding at the Sti1 domain was developed and referred to as the "barcoded binding assay". This assay works by manipulating the molecular weight of each substrate variant with the addition of a variable number of SH3 domains, followed by an IP for UBQLN and resolving the product on a gel to indicate successful binding events. This novel assay is simple in design and interpretation, and is a straightforward and scalable way to systematically probe the characteristics of client-Sti1 interactions. A few clarifying experiments would strengthen the broad potential utility of this tool for other researchers. Further biophysical characterization, using hydropathy scales and Agadir scoring, reveal a set of characteristics that appear to define 'good' UBQLN substrates. The researchers were able to conclude that the Sti1 domain of UBQLNs bind moderately hydrophobic substrates that also have low helicity, prompting the identification of placeholder sequences within the UBQLN sequence itself that bind the Sti1 hydrophobic pocket. Different UBQLNs had unique preferences, which was explored in some depth. To connect to neurodegenerative disease, the researchers showed that common ALS

mutations in UBQLN2 caused varying loss in binding to these placeholder sequences, leading to their final model in the altered phase separation in UBQLNs upon mutation. In short, this is an excellent manuscript that adds significantly to the UBQLN field and needs a few additional clarifying experiments.

Major Comments:

1. While we appreciate the evidence presented that the addition of SH3 domains does not influence the interaction with UBQLNs in Figure 3B, it would strengthen the manuscript to have a control demonstrating more directly that the number of SH3 domains does not influence co-precipitation of the various mutant proteins. For example, rotating/scrambling the number of SH3 domains attached to the TMD sequences that were previously used from Figure 3E and compare the results.
2. A point of confusion in the text of the manuscript is that occasionally it is not clear which UBQLN is being tested or discussed. In addition, there are a few places in the Figures that would benefit from more extensive labeling to clarify which UBQLN is being referenced. For example, Figure 3D should be labeled as "WT UBQLN1" and "Binding Mutant UBQLN1".
3. UBQLN2 is unique for the presence of the PXX domain, which is a mutational hotspot in some cases of familial ALS. The PXX domain is proline-rich and could form a PPII helix, though this may be unlikely given modeling and NMR data (Dao et al. 2018 Mol Cell). However, since PPII helices are known to bind SH3 domains; it would be nice to see a control with dPXX to see if that abrogates Omp25 binding in Figure 3E-F.

Minor Comments:

- Without knowing the (n) or having statistics, it is hard to interpret some of the very valuable quantitations in Figure 2E, 3F, 5D-F, and 6C for example. I'm not sure what statistical test is most acceptable here but at the least showing each replicate as a bar graph with scatter dot plot would be welcome.
- A more in-depth explanation of the Agadir scoring system would be beneficial for the overall understanding of how these values are determined and how they relate to the helical propensity of certain peptides.
- Sti1-II and Sti1-2 are both used to refer to the second STI1 domain in the text
- The authors refer to the specific subfigure (i.e. Figure S1A) the majority of the time, but there are instances where this is not done; I would keep this consistent.
- In paragraph 6 of the results section, there should be a reference to Figure 2D/E.
- In the third paragraph under 'Development of a barcoded binding assay...' in the sentence after Figure 3C is introduced, UBQLN1 is misspelled
- It would be helpful to label the SH3 domains in the schematic of Figure 3A and to include all six plasmid variants.
- When the TMD sequences are included next to the gel (i.e. Figure 3E), it would be helpful to state the barcoding pattern (number?) that each sequence is associated with.
- Figure 4C should specify the UBQLN that is represented in the data (UBQLN1?)
- Figure S2 it would be great to see all UBQLNs aligned with Dsk2 since they are used frequently
- Figure S3A has a misalignment of the labeling or an extra band somewhere
- Figure S6 should specifically label by name the PH sequences in cyan, magenta, and yellow
- There is a period missing in the last sentence of paragraph 5 of the discussion.
- Citations 51 and 52 are the same.
- Figure 6 is very interesting for its examination of the PXX domain as a putative 'fourth' PH, and could be expanded in its discussion. For example, could this explain why UBQLN2 is the most likely to phase separate (Dao et al. Biophysics J 2024)?

Referee #2:

Summary and Significance:

UBQLNs are a family of proteins that play a key role in protein quality control by shuttling ubiquitinated proteins to the proteasome. UBQLNs consist of a UBL, UBA, and two STI1 domains. Mutations to the STI1 domains have been found in patients with amyotrophic lateral sclerosis (ALS). The authors solve the first structure of a STI1-domain from the Ubiquilin class of STI1 domain containing proteins, the Dsk2 STI1-domain bound to substrate. This structure reveals how the STI1 domain folds to form a hydrophobic groove and that it can dimerize and form a hydrophobic chamber, fully enclosing the hydrophobic segment of a substrate. They also demonstrated that ALS mutations disrupt the STI1-substrate binding, revealed that STI1 domain substrates are hydrophobic with low helical propensity, and identified internal regions within UBQLNs that bind to the STI1 domains as placeholder sequences to prevent the exposures of the STI1 domain hydrophobic groove.

Generally, the analyses are interesting and further our understanding of how the STI1-domain in ubiquilins preferentially bind to substrates. Furthermore, the authors identify internal regions within the UBQLNs that are capable of binding to the STI1 domain revealing how the domain remains folded and its hydrophobic groove protected in the absence of substrate.

Major Comments:

1. The authors should make the distinction between the fungal Dsk2 and mammalian UBQLNs as well as when they are referring to each clear.
 - a. In the results section the authors report a crystal structure of the STI1 domain from Dsk2 bound to a substrate. This is the first

time they mention Dsk2. They should introduce Dsk2 as the closest yeast homolog to UBQLNs.

b. Building from this, the authors need to clarify the distinction between Dsk2 and UBQLNs and that they solved the structure of the yeast Dsk2 and that the barcoded binding assay was done with mammalian UBQLNs.

c. Figure 1 appears to discuss the structure of Dsk2-STI1, but includes a domain diagram for UBQLN2 in A. To make it clear that panels C and D are of Dsk2-STI1, they should state so in the figure legends. ie "(C) Surface representation of Dsk2-Sti1 domain, colored by electrostatics. Note how the TMD binds in a hydrophobic groove."

2. Adding a panel with a domain diagram of UBQLN2 with the ALS mutations in Figure 2 such as in Figure 7A, but with less detail. This will help readers follow which STI1 domains these mutants are found in. As Figure 2 reads now, it can be interpreted that one STI1 domain is affected by the Q425R mutant whereas P440L and P189T are found on the same STI1 domain in UBQLN2. Additionally, the authors only test three mutants but mention four. Was there a reason by the Q425R mutant wasn't tested?

3. The main figures are difficult to interpret even with the accompanying figure legends.

a. Figure 3 is difficult to interpret. Panel D is missing labels for the bands, I presume this is the same as in panel E, but it isn't clear. Both panels E and D have more than six bands. Specifically in the Input and FT lanes, what are these bands? Panel F is the quantification of panel E, but it is unclear which TMD sequence in panel E is which barcode, ie 0, 1, 2, etc., in panel F. Panel E should also have a graphic that allows the reader to easily relate the TMD sequence with the gravity score, so either a graphic with the increasing hydrophobicity next to the TMD sequences or the GRAVY score for each sequence. The SDS-PAGE gels should have markers for molecular weight markers.

b. Figure 4 panel D. What do each one of the dots represent? Are these all the variations of OMP25, Sec61 β , Tom5, and VAMP2?

c. Figure 5 - in many of the blots in panel C the FT or Elute band is darker than the Input. This is particularly noticeable in the quantifications of the blots in panel E where the elution/input % is above 100. Why would there be more protein present in the elution than in the input? The authors should address this.

4. In the investigation of STI1 selectivity - The authors explore both TMD hydrophobicity and helical propensity as potential features that drive STI1 selectivity based on a previous report ER TA proteins have both more hydrophobic TMDs and helical propensities than mitochondrial TA proteins (PMID 27925580). While there was no correlation between helicity and UBQLN binding, the authors found that UBQLN substrates are hydrophobic with low helical propensity. A previous study reported ER TA proteins are more amphipathic, having a hydrophobic face within an 11 amino acid segment, than mitochondrial TA proteins (PMID: 34288289). Given that the authors report that PH3 form an amphipathic helix (Fig S7) and this previous report, do the authors see an anti-correlation between a hydrophobic face and UBQLN binding? Could this previously reported feature in ER TA proteins that are missing in mitochondrial TA proteins explain why PH3 has the weakest binding to UBQLN-STI1?

Minor Comments:

1. Sti1 should be capitalized as STI1. The capitalized version refers to the domain whereas Sti1 refers to the protein.

2. The authors interchange between Sti1-1/Sti1-2 and Sti1-I/Sti1-II, the field refers to these domains with the roman numeral versions i.e. STI1-I and STI1-II.

3. "The Omp25 and Sec61b substrate series showed that Ubiquilins bound preferentially to TMDs with moderate hydrophobicity (Figure 3E-G & S3)"

Referee #3:

This manuscript describes the crystal structure of a yeast Dsk2 Sti1 domain bound to a variant of the VAMP2 transmembrane domain and uses a barcoded binding assay to study interactions between human UBQLN proteins and TMD variants. The crystal structure shows a Sti1 dimer with the TMD buried in an internal cavity. Enthusiasm would be higher if findings from this structure were tested directly in the human UBQLN proteins to determine whether this dimerization mechanism is conserved. For example, do the two Sti1 domains of the human protein associate like the yeast dimer? Another strength is that the authors use the PURExpress IVT system to evaluate TMD binding to UBQLN proteins, including ALS Sti1 mutations. They identify sequences in UBQLN that bind to the Sti1 domain, including the PXX repeat. A strength of these experiments is the finding that the UBQLN2 P189T mutation prevents TMD and PXX repeat binding, suggesting that these interactions occur at Sti1-1. Enthusiasm would be higher if these results were tested in human cells (such as by immunoprecipitation experiments) and following further characterization of the P189T mutation to rule out technical effects - such as its own aggregation. Another aspect is that the binding experiments seem to be done with full length UBQLN raising the possibility that other domains/parts of the sequence contribute to the observations. More direct evidence for the PXX repeat or other internal sequence interaction with the Sti1 domain would elevate the manuscript. Nonetheless, the manuscript has compelling strengths, with new insights regarding the impact of ALS mutations. Some additional suggestions are made below.

PXX should be defined as being PXX repeats.

'moderately hydrophobic transmembrane domains' is more of a property than 'motif' but also it'd be helpful to specify the differentiation - is it that most other TMDs are 'exclusively' hydrophobic? Maybe some more information is needed here. It seems that 'exclusively hydrophobic' would be a property of single pass helices but certainly not ion channels for example. Again, this part is unclear.

Figure 1A: Sti1-1 and Sti1-2 should be explicitly defined and terminal amino acids included. In addition the region crystalized in

panel B should be boxed or indicated in some way indicated in a domain layout for *M. bicupsidata* Dsk2.

Figure 1B - D: The Sti1 and TMDs should be labeled in the figure for clarity.

The reference to the size exclusion chromatography data should be Figure S1C for clarity. The molecular weight markers suggest that the crystallization construct runs as a trimer; is this correct? The caption indicates that it has a molecular weight of 24.4 kDa. Is so how does this finding impact the interpretation of the crystal structure, which shows a dimer?

A figure should be included that shows the 'key residues' and their interactions as mentioned in the text.

More information should be given on the dimerization mechanism by showing amino acids at the interface. In addition, the terminal ends of each monomer should be clearly labeled. What role does the TMD have in driving dimerization? For example if TMD is removed from the crystallization construct does size exclusion chromatography indicate an oligomeric molecular weight? Is there any evidence for the human Sti1 domain forming a similar dimeric structure and can some experimental evidence be obtained to support similarity between the yeast structure and what happens in the human protein?

Figure S2: only UBQLN2 is shown whereas the text indicates UBQLN1/4 are also shown and there's a typo 'UBLQN'.

Figure 2A/B: The nearby amino acids in TMD should be shown and molecular dynamics run to evaluate the likely impact of the Q425R and M446R mutations. The predicted AlphaFold structures referred to at the bottom of p. 4 should be shown.

Figure 2C: The predicted structural disruption should be tested experimentally or at least with molecular dynamics.

p. 5: Figure 2D should be referenced here: "We observed robust binding of Omp25 to WT UBQLN2 and no binding in the absence of Ubiquilins." Figure 2E should be referenced two sentences later.

Figure 2D: More information is needed for this experiment in the Results section and figure caption. Information is provided in Methods but for example, adding 'elution with 3x Flag peptide' would make the manuscript easier to follow and adds very little additional text. The membrane should be reprobed with anti-FLAG antibodies to evaluate levels of UBQLN. Molecular weight markers are needed. As is, it would appear that P440L and M446R bind more strongly to Omp25. Statistical analyses should be done.

Figure 3: the subpanels need to be individually referenced where appropriate. For example, on p. 8, first full paragraph, last two sentences: which panels should readers be looking at for this? Where I'm looking 1.61 to 3.29 seems incorrect (Figure 3G).

Where is this data "The Omp25 and Sec61b substrate series showed that Ubiquilins bound preferentially to TMDs with moderate hydrophobicity (Figure 3 & S3)." Figure 3G should be referenced.

GRAVY score needs to be properly defined when first introduced, for example as an overall hydrophobicity value for the TMD region - is this even correct? How do the data in Figure 3G relate to the sequences in Figure 3E? The barcode values need to be listed for each sequence.

It's written that Vamp2 shows 'no clear up and down pattern'. There is a pattern correlating increased GRAVY score with decreased elution/input however.

The statement "Having determined the biophysical parameters that govern substrate binding to the Sti1 domains" on p.10 is an overstatement and should be either made more specific or simply deleted.

Why is AlphaFold2 being used instead of AlphaFold3? A better program for disordered sequences might be IDPConformerGenerator.

Labels need to be included on Figure 5A indicating Sti1-1, etc.

Figure 5C: UBQLN2 needs to be immunoprobed. Is this assay done with the full length protein?

This interpretation is not strictly correct "indicating that Sti1 domain interaction with placeholder sequences is conserved from yeast to humans (Figure 5C)." The loss of binding to P189T, which is in the Sti1 domain, written about later provides evidence for this interpretation. Also it would seem that binding to P189T is not statistically greater than background, making 'partial loss' not quite correct. Statistical analyses should be done for these data. If PH1 can bind to Sti1-2, then why is all binding lost for P189T?

It's not clear what the authors mean by "examination of the Agadir and GRAVY scores shows biophysical parameters consistent with the other placeholder sequences identified above (Figure 6A)." The GRAVY and Agadir scores in Figure 6A do not indicate consistency with Figure 4D.

More characterization should be done on UBQLN2 P189T, since it is the consistent non-binder from which binding to Sti1-1 is proposed.

There is no figure showing the 'methionine rich hydrophobic groove'.

Referee #1:

Onwunma et al. is an elegant and thought-provoking biochemical investigation into UBQLN function through a very thorough characterization of the interactions of the Sti1 domain. The authors obtained a crystal structure of the yeast Dsk2 Sti1 domain bound to a transmembrane domain (TMD) at 1.98 Å, showing dimerization that creates a hydrophobic groove for TMD binding. A biochemical assay to probe the characteristics of TMD binding at the Sti1 domain was developed and referred to as the "barcoded binding assay". This assay works by manipulating the molecular weight of each substrate variant with the addition of a variable number of SH3 domains, followed by an IP for UBQLN and resolving the product on a gel to indicate successful binding events. This novel assay is simple in design and interpretation, and is a straightforward and scalable way to systematically probe the characteristics of client-Sti1 interactions. A few clarifying experiments would strengthen the broad potential utility of this tool for other researchers. Further biophysical characterization, using hydropathy scales and Agadir scoring, reveal a set of characteristics that appear to define 'good' UBQLN substrates. The researchers were able to conclude that the Sti1 domain of UBQLNs bind moderately hydrophobic substrates that also have low helicity, prompting the identification of placeholder sequences within the UBQLN sequence itself that bind the Sti1 hydrophobic pocket. Different UBQLNs had unique preferences, which was explored in some depth. To connect to neurodegenerative disease, the researchers showed that common ALS mutations in UBQLN2 caused varying loss in binding to these placeholder sequences, leading to their final model in the altered phase separation in UBQLNs upon mutation. In short, this is an excellent manuscript that adds significantly to the UBQLN field and needs a few additional clarifying experiments.

Major Comments:

1. While we appreciate the evidence presented that the addition of SH3 domains does not influence the interaction with UBQLNs in Figure 3B, it would strengthen the manuscript to have a control demonstrating more directly that the number of SH3 domains does not influence co-precipitation of the various mutant proteins. For example, rotating/scrambling the number of SH3 domains attached to the TMD sequences that were previously used from Figure 3E and compare the results.

We agree that it is important to show that the SH3 domains do not unduly influence substrate binding to Ubiquilins. We therefore designed a substrate series where we have the same TMD for all six bar codes (Figure S5B & C). As expected, there is no significant difference in Ubiquilin binding.

2. A point of confusion in the text of the manuscript is that occasionally it is not clear which UBQLN is being tested or discussed. In addition, there are a few places in the Figures that

would benefit from more extensive labeling to clarify which UBQLN is being referenced. For example, Figure 3D should be labeled as "WT UBQLN1" and "Binding Mutant UBQLN1".

Thank you for pointing this out. We have revised the text and figures throughout the manuscript to clarify which Ubiquilin paralog is being referenced. We have revised the labeling in Figure S5A, which was Figure 3D in the original manuscript.

3. UBQLN2 is unique for the presence of the PXX domain, which is a mutational hotspot in some cases of familial ALS. The PXX domain is proline-rich and could form a PPII helix, though this may be unlikely given modeling and NMR data (Dao et al. 2018 Mol Cell). However, since PPII helices are known to bind SH3 domains; it would be nice to see a control with dPXX to see if that abrogates Omp25 binding in Figure 3E-F.

We agree that we need to explicitly rule out the potential of the PXX domain interacting with SH3 domains. As suggested, we generated a Δ PXX UBQLN2 variant and repeated the binding assay (Figure S5D & E). We observed no major difference in substrate binding. We would also like to point out that the barcoded binding assay results with UBQLN1 and UBQLN4, which lack the PXX domain, largely follow the same trend as UBQLN2. Collectively, this argues that there is no significant PXX-SH3 interaction.

Minor Comments:

- Without knowing the (n) or having statistics, it is hard to interpret some of the very valuable quantitations in Figure 2E, 3F, 5D-F, and 6C for example. I'm not sure what statistical test is most acceptable here but at the least showing each replicate as a bar graph with scatter dot plot would be welcome.

We have replotted all graphs as bar graphs with scatter dot plots and performed statistical analysis. Due to the density of information and number of potential comparisons in Figure 3F, we chose not to display statistical comparisons.

- A more in-depth explanation of the Agadir scoring system would be beneficial for the overall understanding of how these values are determined and how they relate to the helical propensity of certain peptides.

Thanks for this helpful suggestion. We have added more background on the Agadir score.

- Sti1-II and Sti1-2 are both used to refer to the second STI1 domain in the text

Thank you for pointing this out, we have standardized the nomenclature to STI1-II.

- The authors refer to the specific subfigure (i.e. Figure S1A) the majority of the time, but there are instances where this is not done; I would keep this consistent.

We now refer to specific subfigures.

- In paragraph 6 of the results section, there should be a reference to Figure 2D/E.

We changed this

- In the third paragraph under 'Development of a barcoded binding assay...' in the sentence after Figure 3C is introduced, UBQLN1 is misspelled

Thanks for pointing out this typo, we corrected it.

- It would be helpful to label the SH3 domains in the schematic of Figure 3A and to include all six plasmid variants.

Thanks for helping improve our figure clarity, we changed this.

- When the TMD sequences are included next to the gel (i.e. Figure 3E), it would be helpful to state the barcoding pattern (number?) that each sequence is associated with.

We changed this

- Figure 4C should specify the UBQLN that is represented in the data (UBQLN1?)

The data is similar for all Ubiquilins. The original draft of the manuscript used UBQLN1, but we decided to include UBQLN2 in the revised version as it makes the manuscript more cohesive.

- Figure S2 it would be great to see all UBQLNs aligned with Dsk2 since they are used frequently

We made this change.

- Figure S3A has a misalignment of the labeling or an extra band somewhere

We have now updated all figures with the barcoded binding assay to clearly label the barcode and TMD sequence.

- Figure S6 should specifically label by name the PH sequences in cyan, magenta, and yellow

We made this change

- There is a period missing in the last sentence of paragraph 5 of the discussion.

We changed this.

- Citations 51 and 52 are the same.

Good catch. We fixed this.

- Figure 6 is very interesting for its examination of the PXX domain as a putative 'fourth' PH, and could be expanded in its discussion. For example, could this explain why UBQLN2 is the most likely to phase separate (Dao et al. Biophysics J 2024)?

This is a great suggestion. We have updated the discussion section to reflect this possibility.

Referee #2:

Summary and Significance:

UBQLNs are a family of proteins that play a key role in protein quality control by shuttling ubiquitinated proteins to the proteasome. UBQLNs consist of a UBL, UBA, and two STI1 domains. Mutations to the STI1 domains have been found in patients with amyotrophic lateral sclerosis (ALS). The authors solve the first structure of a STI1-domain from the Ubiquilin class of STI1 domain containing proteins, the Dsk2 STI1-domain bound to substrate. This structure reveals how the STI1 domain folds to form a hydrophobic groove and that it can dimerize and form a hydrophobic chamber, fully enclosing the hydrophobic segment of a substrate. They also demonstrated that ALS mutations disrupt the STI1-substrate binding, revealed that STI1 domain substrates are hydrophobic with low helical propensity, and identified internal regions within UBQLNs that bind to the STI1 domains as placeholder sequences to prevent the exposures of the STI1 domain hydrophobic groove.

Generally, the analyses are interesting and further our understanding of how the STI1-domain in ubiquilins preferentially bind to substrates. Furthermore, the authors identify internal regions within the UBQLNs that are capable of binding to the STI1 domain revealing how the domain remains folded and its hydrophobic groove protected in the absence of substrate.

Major Comments:

1. The authors should make the distinction between the fungal Dsk2 and mammalian UBQLNs as well as when they are referring to each clear.

a. In the results section the authors report a crystal structure of the STI1 domain from Dsk2 bound to a substrate. This is the first time they mention Dsk2. They should introduce Dsk2 as the closest yeast homolog to UBQLNs.

Thanks for pointing this out. We now introduce Dsk2 in the 3rd paragraph of the introduction.

b. Building from this, the authors need to clarify the distinction between Dsk2 and UBQLNs and that they solved the structure of the yeast Dsk2 and that the barcoded binding assay was done with mammalian UBQLNs.

We have made several changes throughout the manuscript and figure legends to better emphasize that we solved the crystal structure of the yeast Dsk2 STI1 domain whereas the binding assays were performed with human Ubiquilins.

c. Figure 1 appears to discuss the structure of Dsk2-STI1, but includes a domain diagram for UBQLN2 in A. To make it clear that panels C and D are of Dsk2-STI1, they should state so in the figure legends. ie "C) Surface representation of Dsk2-Sti1 domain, colored by electrostatics. Note how the TMD binds in a hydrophobic groove."

This is a helpful suggestion. We have made these changes.

2. Adding a panel with a domain diagram of UBQLN2 with the ALS mutations in Figure 2 such as in Figure 7A, but with less detail. This will help readers follow which STI1 domains these mutants are found in. As Figure 2 reads now, it can be interpreted that one STI1 domain is affected by the Q425R mutant where as P440L and P189T are found on the same STI1 domain in UBQLN2. Additionally, the authors only test three mutants but mention four. Was there a reason by the Q425R mutant wasn't tested?

Thank you for this helpful suggestion, we have now remade figure 2 to more clearly show which mutations are in the STI1-I and STI1-II domains.

We understand that it may appear incomplete to not include the Q425R mutation. However, we don't believe making this mutation is necessary for the story. The new co-IP experiments in Figure 2I show that the Q425R mutation is the least disruptive mutation to Omp25 binding. As our structural analysis also predicts that the Q425R mutation will be the less severe than the P440L or M446R mutation, we felt that generating this mutant would yield no additional insights.

3. The main figures are difficult to interpret even with the accompanying figure legends. a. Figure 3 is difficult to interpret. Panel D is missing labels for the bands, I presume this is the same as in panel E, but it isn't clear. Both panels E and D have more than six bands. Specifically in the Input and FT lanes, what are these bands? Panel F is the quantification of panel E, but it is unclear which TMD sequence in panel E is which barcode, ie 0, 1, 2, etc., in panel F. Panel E should also have a graphic that allows the reader to easily relate the TMD sequence with the gravity score, so either a graphic with the increasing hydrophobicity next to the TMD sequences or the GRAVY score for each sequence. The SDS-PAGE gels should have markers for molecular weight markers.

Thank you for this suggestion. We have now added extensive labeling for figure 3 and supplemental figures S6, 7, & 8 to allow the reader to easily see the hydrophobicity and barcode across all figure subpanels. We labeled the barcode on relevant bands, added molecular weight markers, and provided the TMD sequence where appropriate. We also clarify which TMD series is used in the text and figure legends.

We consistently see expression of some non-specific bands in our assays. Because we perform the barcoded binding assay by adding full-length plasmid, we suspect that

this may arise from transcription of other open reading frames in the plasmids. To help the reader focus on the relevant bands, we now label the barcode for each band.

b. Figure 4 panel D. What do each one of the dots represent? Are these all the variations of OMP25, Sec61 β , Tom5, and VAMP2?

We apologize for the lack of clarity. Yes, each spot represents a different TMD variant of OMP25, Sec61 β , Tom5, and VAMP2 from the four barcoded binding assays in figures 3, S6, S7, and S8. The size of each spot corresponds to how well each substrate binds to UBQLN2. We have modified the text to better clarify this.

c. Figure 5 - in many of the blots in panel C the FT or Elute band is darker than the Input. This is particularly noticeable in the quantifications of the blots in panel E where the elution/input % is above 100. Why would there be more protein present in the elution than in the input? The authors should address this.

We apologize for the confusion. As noted in the methods, the Elution fraction is 5x more concentrated than the Input and FT fractions. We found that we could get more accurate and reproducible quantitation if all lanes had roughly equal band intensity. This change was applied uniformly across all samples.

We suspect that the enhanced intensity of the FT band compared to the Input was due to pipetting error. To test for this, we repeated all experiments in Figure 5 and now have better representative images and more robust statistics for our binding assay.

4. In the investigation of STI1 selectivity - The authors explore both TMD hydrophobicity and helical propensity as potential features that drive STI1 selectivity based on a previous report ER TA proteins have both more hydrophobic TMDs and helical propensities than mitochondrial TA proteins (PMID 27925580). While there was no correlation between helicity and UBQLN binding, the authors found that UBQLN substrates are hydrophobic with low helical propensity. A previous study reported ER TA proteins are more amphipathic, having a hydrophobic face within an 11 amino acid segment, than mitochondrial TA proteins (PMID: 34288289). Given that the authors report that PH3 form an amphipathic helix (Fig S7) and this previous report, do the authors see an anti-correlation between a hydrophobic face and UBQLN binding? Could this previously reported feature in ER TA proteins that are missing in mitochondrial TA proteins explain why PH3 has the weakest binding to UBQLN-STI1?

This is an interesting suggestion. During our initial analysis we looked at the formation of an amphipathic helix and saw no clear correlation or anti-correlation with substrate binding, which is why it was excluded from the initial draft of the manuscript. Our analysis is below. Briefly, we took the best and worst binding

substrates from the Omp25, Sec61 β , VAMP2, and TOM5 substrate series and used the Wheel Face and Patch methods to identify residues that make an amphipathic helix as indicated in PMID: 34288289. We then measured the hydrophobicity of the hydrophobic face using GRAVY, which is based on the Kyte-Doolittle scale. As shown in Response Figure 1A, there is no clear pattern between the hydrophobicity of the hydrophobic face and binding to UBQLN2.

We then asked if the difference in hydrophobicity of the two faces of the amphipathic helix can predict substrate binding. Here, we calculated the GRAVY score of the hydrophobic and hydrophilic faces of the amphipathic helix. We then plotted substrate binding as a function of the differences in GRAVY scores (Response Figure 1B). Again, there is no clear pattern.

We agree with the reviewer that a thorough examination of STI1 domain interaction with amphipathic helices is an important area for future investigation, particularly as our crystal structure shows that a single STI1 domain (as opposed to the dimer) could accommodate an amphipathic helix. We now address this in the discussion section and include a reference to PMID: 34288289.

Response Figure 1: Amphipathic helix

- A) The GRAVY score (hydrophobicity) of the hydrophobic face of an amphipathic helix does not predict substrate binding to UBQLN2.
- B) The difference in GRAVY scores between the hydrophobic and hydrophilic faces of an amphipathic helix does not predict substrate binding to UBQLN2.

Minor Comments:

1. Sti1 should be capitalized as STI1. The capitalized version refers to the domain whereas Sti1 refers to the protein.

We made this change.

2.The authors interchange between Sti1-1/Sti1-2 and Sti1-I/Sti1-II, the field refers to these domains with the roman numeral versions i.e. STI1-I and STI1-II.

We made this change.

3."The Omp25 and Sec61b substrate series showed that Ubiquilins bound preferentially to TMDs with moderate hydrophobicity (Figure 3E-G & S3)"

Sorry, we are a bit confused on what needs to be changed here as this comment only shows a direct quotation from our original manuscript.

Referee #3:

This manuscript describes the crystal structure of a yeast Dsk2 Sti1 domain bound to a variant of the VAMP2 transmembrane domain and uses a barcoded binding assay to study interactions between human UBQLN proteins and TMD variants. The crystal structure shows a Sti1 dimer with the TMD buried in an internal cavity. Enthusiasm would be higher if findings from this structure were tested directly in the human UBQLN proteins to determine whether this dimerization mechanism is conserved. For example, do the two Sti1 domains of the human protein associate like the yeast dimer?

We agree with the reviewer that an exciting potential insight from our manuscript is the ability to provide structural insights into UBQLN2 oligomerization. Indeed, previous work has shown that the STI-II domain of human UBQLN2 is essential for dimerization. This is shown in Figure S2 from Dao et al., 2024 *Biophysical Journal* (see below). We modified the text to better emphasize this relationship.

Figure S2 from Dao et al., 2024 Biophysical Journal not shown

The data on Dsk2 is less clear cut, so we have taken a cautious approach so as not to overinterpret our data. This manuscript was co-submitted with a manuscript by Acharya et al., which examines the STI1 domain in *S. cerevisiae* Dsk2 (<https://www.biorxiv.org/content/10.1101/2025.03.14.643327v1>). In that manuscript, the authors demonstrate that while there appears to be some transient interaction between STI1 domains, *S. cerevisiae* Dsk2 does not form stable dimers.

As suggested below, we examined the contribution of the TMD to the oligomerization of the crystal construct. We generated a version of our crystal construct lacking the TMD (STI1-FrzS- Δ TMD) and performed size exclusion chromatography. This construct eluted significantly later, indicating that the TMD is a major contributor to the oligomerization of the crystal construct. The apparent molecular weight of the STI1-FrzS- Δ TMD construct was 45 kDa, which corresponds to a dimer (Figure S2B).

To test if the STI1-FrzS- Δ TMD construct forms a stable dimer, we sought to disrupt STI1-STI1 contacts. We identified residues M29, N37, and M71 as key regulators of the STI1-STI1 dimer interface in our crystal structure (Figure S2D). These residues correspond to amino acids M189, N197, and M231 in the *M. biscaupidata* Dsk2. We generated the M29D, N37A, M71D triple point mutation in the STI1-FrzS- Δ TMD construct and performed size exclusion chromatography. We observed a modest shift in elution volume, but the construct still did not elute at the volume expected for a monomer (Figure S2B). The most parsimonious explanation for these results is that the crystallization construct runs anomalously large on size exclusion chromatography due to an extended conformation.

In light of these results, it is unclear if the UBQLN2 STI1-II forms a stable dimer by itself or if dimerization is driven by the STI1-placeholder interaction. Differentiating between these models will require extensive work that we feel is beyond the scope of this manuscript. We therefore limited our conclusions to simply state that the STI1-TMD interaction is a major driver of oligomerization in the crystal construct. We also use the discussion section to point out that more work is needed for a full structural understanding of the STI1 dimerization in human Ubiquilins.

Another strength is that the authors use the PURExpress IVT system to evaluate TMD binding to UBQLN proteins, including ALS Sti1 mutations. They identify sequences in UBQLN that bind to the Sti1 domain, including the PXX repeat. A strength of these experiments is the finding that the UBQLN2 P189T mutation prevents TMD and PXX repeat binding, suggesting that these interactions occur at Sti1-1. Enthusiasm would be higher if these results were tested in human cells (such as by immunoprecipitation experiments) and following further characterization of the P189T mutation to rule out technical effects - such as its own aggregation.

Thank you for this suggestion. We agree that given the consistently strong effect observed with the P189T mutation that it is important to rule out alternative models, such as the P189T mutation being aggregation prone. Accordingly, we performed size exclusion chromatography with the P189T mutant to show that the peak perfectly overlaps with WT UBQLN2 (Figure S4A). We also performed an IP with all UBQLN2 point mutants to show that there is no defect in IP efficiency (Figure S4B).

We hypothesized that Omp25 binds to both STI1 domains but has higher affinity for the STI1-I domain and the concentrations of Omp25 and/or UBQLN2 are such that Omp25 predominantly binds to the STI1-I domain under our assay conditions. To test this hypothesis, we repeated the binding assay with 23.5 μ M UBQLN2 instead of 3 μ M, with the expectation that the higher concentration of UBQLN2 would allow Omp25 to bind to the STI1-II domain. Consistent with our hypothesis, we observed that Omp25 now binds to both WT and P189T UBQLN2 (Figure S4C & D). We attempted to further this analysis by generating constructs containing mutations in both STI1 domains (P189T/Q425R, P189T/P440L, and P189T/M446R). Despite numerous attempts, we ran into technical issues and were unable to purify UBQLN2 double mutants at sufficient concentration to perform the in vitro binding assay with 23.5 mM UBQLN2.

As suggested, we attempted to show the interaction of UBQLN2 with Omp25 in cells via co-IP, but were unable to see interaction under standard IP conditions (Response Figure 2A). The interaction of Ubiquilins with Omp25 is well documented in the literature (Itakura et al. *Molecular Cell* 2016, Guna et al. *Science* 2022). Previous work has shown that the STI1-Omp25 interaction is transient (Itakura et al *Molecular Cell*

2016). Furthermore, our work shows that this interaction is driven by hydrophobic interactions. We hypothesized that the STI1-Omp25 interaction was sensitive to detergents used in our IP buffers.

We next attempted to stabilize the UBQLN2-Omp25 interaction by cross-linking IP. Importantly, the cross-linking occurs before the cells are exposed to detergents during cell lysis. This approach was effective in stabilizing the UBQLN2-Omp25 interaction (Response Figure 2B). Consistent with our use of the strong CMV promoter leading to Omp25 and UBQLN2 overexpression, there was minimal difference in binding between the WT and P189T mutant,

We hypothesized that the overexpression of UBQLN2 and Omp25 in the co-IP could allow for Omp25 binding to the STI1-II domain. To test this hypothesis, we repeated the co-IP experiment with the P189T/Q425R, P189T/P440L, and P189T/M446R double mutants, which have mutations in both STI1 domains. Consistent with our hypothesis, all double mutants showed a defect in Omp25 binding (Figure 2I).

In conclusion, we ruled out several trivial reasons for the lack of substrate binding to the P189T mutant. Instead, we show that substrates bind to both STI1 domains but have higher affinity for the STI1-I domain. We confirmed that Omp25-UBQLN2 interaction in cells via cross-linking co-IP and showed that making constructs with mutations in both STI1 domains can disrupt substrate binding.

Response Figure 2: Interaction of UBQLN2 with Omp25 in cells requires DSP cross-linking before IP.

- A) FLAG-tagged Omp25 and HA-tagged UBQLN2 were co-transfected into WT HEK cells. Anti-FLAG IP shows no detectable UBQLN2 in the elution fraction.
- B) As in A, but samples were cross-linked with 1 mM DSP for 30 minutes prior to cell lysis and immunoprecipitation. Both WT and P189T UBQLN2 are detectable in the elution fraction.

Another aspect is that the binding experiments seem to be done with full length UBQLN raising the possibility that other domains/parts of the sequence contribute to the observations. More direct evidence for the PXX repeat or other internal sequence interaction with the Sti1 domain would elevate the manuscript. Nonetheless, the manuscript has compelling strengths, with new insights regarding the impact of ALS mutations. Some additional suggestions are made below.

This is a great point and we apologize for the lack of clarity. This manuscript was co-submitted with a manuscript by Acharya et al., which examines the STI1 domain in *S. cerevisiae* Dsk2. In that manuscript, the authors used NMR to identify the placeholder sequences and show direct interaction of these sequences with the STI1 domain. Furthermore, previous work (Dao et al. *Structure* 2019) has shown that the PXX region can interact with the STI1 domain. We believe that the NMR studies from Acharya et al. combined with our binding assay robustly show the interaction of internal sequences with the STI1 domain. However, the reviewer's point is well taken and we have modified the text to acknowledge the possibility that these internal placeholder sequences also interact with other Ubiquilin domains.

PXX should be defined as being PXX repeats.

We made this change

'moderately hydrophobic transmembrane domains' is more of a property than 'motif' but also it'd be helpful to specify the differentiation - is it that most other TMDs are 'exclusively' hydrophobic? Maybe some more information is needed here. It seems that 'exclusively hydrophobic' would be a property of single pass helices but certainly not ion channels for example. Again, this part is unclear.

The original analysis was carried out on single pass TMDs, which were all "exclusively hydrophobic". However, the overall hydrophobicity within the TMD can vary based on the ratio of moderately hydrophobic residues (Alanine) with high hydrophobic residues (Leucine). We modified the text to clarify this. We also changed "motif" to "property".

Figure 1A: Sti1-1 and Sti1-2 should be explicitly defined and terminal amino acids included.
We updated figure 1A to explicitly define the residues in the STI domains

In addition the region crystalized in panel B should be boxed or indicated in some way indicated in a domain layout for *M. bicuspidata* Dsk2.

We labeled the domain boundaries in Figure 1A and added text to the figure legend to indicate which residues were included in the crystallization construct.

Figure 1B - D: The Sti1 and TMDs should be labeled in the figure for clarity.

We made this change

The reference to the size exclusion chromatography data should be Figure S1C for clarity.

We made this change

The molecular weight markers suggest that the crystallization construct runs as a trimer; is this correct? The caption indicates that it has a molecular weight of 24.4 kDa. Is so how does this finding impact the interpretation of the crystal structure, which shows a dimer?

This is a great question. We believe that this observation supports our model of the placeholder-STI1 interaction being a major driver of multivalency in Ubiquilin oligomerization. As described above, we performed size exclusion chromatography on a Δ TMD version of our crystal construct (STI1-FrzS- Δ TMD). This construct eluted significantly later, indicating that the TMD is a major contributor to the oligomerization of the crystal construct. The apparent molecular weight of the STI1-FrzS- Δ TMD construct was 45 kDa, which corresponds to a dimer (Figure S2B).

To test if the STI1-FrzS- Δ TMD construct forms a stable dimer, we sought to disrupt STI1-STI1 contacts. We identified residues M29, N37, and M71 as key regulators of the STI1-STI1 dimer interface in our crystal structure (Figure S2D). These residues correspond to amino acids M189, N197, and M231 in the *M. bicuspidata* Dsk2. We generated the M29D, N37A, M71D triple point mutation in the STI1-FrzS- Δ TMD construct and performed size exclusion chromatography. We observed a modest shift in elution volume, but the construct still did not elute at the volume expected for a monomer (Figure S2B). The most parsimonious explanation for these results is that the crystallization construct runs anomalously large on size exclusion chromatography due to an extended conformation. We therefore limited our conclusions to simply state that the STI1-TMD interaction is a major driver of oligomerization in the crystal construct.

A figure should be included that shows the 'key residues' and their interactions as mentioned in the text.

We added Figure S1C to explicitly show the key residues that interact with the TMD.

More information should be given on the dimerization mechanism by showing amino acids at the interface. In addition, the terminal ends of each monomer should be clearly labeled. What role does the TMD have in driving dimerization? For example if TMD is removed from the crystallization construct does size exclusion chromatography indicate an oligomeric molecular weight?

As described below, we performed size exclusion chromatography on a Δ TMD version of our crystal construct (STI1-FrzS- Δ TMD). This construct eluted significantly later, indicating that the TMD is a major contributor to the oligomerization of the crystal construct. The apparent molecular weight of the STI1-FrzS- Δ TMD construct was 45 kDa, which corresponds to a dimer (Figure S2B).

To test if the STI1-FrzS- Δ TMD construct forms a stable dimer, we sought to disrupt STI1-STI1 contacts. We identified residues M29, N37, and M71 as key regulators of the STI1-STI1 dimer interface in our crystal structure (Figure S2D). These residues correspond to amino acids M189, N197, and M231 in the *M. biscaupidata* Dsk2. We generated the M29D, N37A, M71D triple point mutation in the STI1-FrzS- Δ TMD construct and performed size exclusion chromatography. We observed a modest shift in elution volume, but the construct still did not elute at the volume expected for a monomer (Figure S2B). The most parsimonious explanation for these results is that the crystallization construct runs anomalously large on size exclusion chromatography due to an extended conformation.

Is there any evidence for the human Sti1 domain forming a similar dimeric structure and can some experimental evidence be obtained to support similarity between the yeast structure and what happens in the human protein?

This is an excellent point. The STI1-II domain of human UBQLN2 is essential for UBQLN2 dimerization. This is shown in Figure S2 from Dao et al., 2024 *Biophysical Journal* (see below). This manuscript was co-submitted with a manuscript by Acharya et al., which examines the STI1 domain in *S. cerevisiae* Dsk2. In that manuscript, the authors demonstrate that while there appears to be some transient interaction between STI1 domains, *S. cerevisiae* Dsk2 does not form stable dimers. As described above, we also performed SEC analysis with a Δ TMD version of our crystal construct and show that the TMD is a major driver of oligomerization in our crystal construct. It is therefore unclear if the UBQLN2 STI1-II domain forms a stable dimer by itself or if dimerization is driven by the STI1-placeholder interaction. Differentiating between these models will require extensive work that we feel is beyond the scope of this manuscript. As described above, we briefly discuss the potential of our structure to inform on Ubiquilin dimerization mechanisms, while also acknowledging the

limitations of our structure due to species-specific differences and the potential role of the TMD in driving oligomerization of the crystal construct. We also use the discussion section to point out that more work is needed for a full structural understanding of the ST11 dimerization in human Ubiquilins.

Figure S2 from Dao et al., 2024 Biophysical Journal not shown

Figure S2: only UBQLN2 is shown whereas the text indicates UBQLN1/4 are also shown and there's a typo 'UBLQN'.

Thanks for catching this. We fixed these errors.

Figure 2A/B: The nearby amino acids in TMD should be shown and molecular dynamics run to evaluate the likely impact of the Q425R and M446R mutations. The predicted AlphaFold structures referred to at the bottom of p. 4 should be shown.

Thank you for this suggestion. We added the AlphaFold3 models of the UBQLN2 ST11-II domains showing that the Q425R and M446R mutations insert a charged residue into the ST11 hydrophobic groove (Figure S3E & F). We also added Figure S3C & D to explicitly show the interactions of Q425 and M446 equivalent residues with the TMD in our structure.

Our lab lacks expertise in molecular dynamics, so we spoke to several experts in the field. They uniformly advised us that addressing these questions via molecular dynamics will be an arduous process that requires atomistic-based simulations rather than coarse grained models. In light of these challenges, we believe that MD simulations are outside the scope of this manuscript.

To acknowledge the reviewer's point, we have modified the text to state that the ALS mutations are predicted to disrupt the structure and/or dynamics of the ST11 hydrophobic groove.

Figure 2C: The predicted structural disruption should be tested experimentally or at least with molecular dynamics.

The reviewer's point is well taken. We have modified the text to state that the ALS mutations are predicted to disrupt the structure and/or dynamics of the ST11 hydrophobic groove.

As discussed above, molecular dynamics will be an arduous process that requires atomistic-based simulations rather than coarse grained models. Similarly, the NMR experiments required to test the structural and/or dynamics changes of the P189T and P440L mutations are beyond the scope of the manuscript.

p. 5: Figure 2D should be referenced here: "We observed robust binding of Omp25 to WT UBQLN2 and no binding in the absence of Ubiquilins." Figure 2E should be referenced two sentences later.

We made this change

Figure 2D: More information is needed for this experiment in the Results section and figure caption. Information is provided in Methods but for example, adding 'elution with 3x Flag peptide' would make the manuscript easier to follow and adds very little additional text.

Thank you for this suggestion, we modified the text and figure legend to clarify that Ubiquilin-substrate complex formation is evaluated by an anti-FLAG IP and that Ubiquilins were eluted with 3x FLAG peptide..

The membrane should be reprobed with anti-FLAG antibodies to evaluate levels of UBQLN.

We agree that our assay depends on having equal input and IP efficiency for all Ubiquilin variants. Unfortunately, we cannot reprobe the blot as suggested. The radioactive substrate is visualized by drying the gel and then exposing it to film. To clarify, the binding assay is performed by *adding equal amounts* of recombinantly purified Ubiquilin to each IVT reaction, so the concentration of Ubiquilin is consistent across all samples. It is formally possible that some of the Ubiquilin variants could have altered IP efficiency. To rule out this possibility, we added a representative anti-FLAG western blot from an IVT assay (Figure S4B). This western blot was performed

exactly as we perform the binding assay in Figure 2F, except that we performed an anti-FLAG western blot rather than drying the gel and performing autoradiography.

Molecular weight markers are needed. As is, it would appear that P440L and M446R bind more strongly to Omp25. Statistical analyses should be done.

We have added molecular weight markers, updated the figure to show individual data points, and performed statistical analysis to demonstrate that there is no significant difference between WT, P440L, and M446R.

Figure 3: the subpanels need to be individually referenced where appropriate. For example, on p. 8, first full paragraph, last two sentences: which panels should readers be looking at for this? Where I'm looking 1.61 to 3.29 seems incorrect (Figure 3G). Where is this data "The Omp25 and Sec61b substrate series showed that Ubiquilins bound preferentially to TMDs with moderate hydrophobicity (Figure 3 & S3)." Figure 3G should be referenced.

We now call out individual figure subpanels throughout the text.

GRAVY score needs to be properly defined when first introduced, for example as an overall hydrophobicity value for the TMD region - is this even correct? How do the data in Figure 3G relate to the sequences in Figure 3E? The barcode values need to be listed for each sequence.

Thank you for this suggestion. We modified the text as follows "we used the Grand Average of Hydrophobicity (GRAVY) score. This simple metric reflects the average hydrophobicity of an amino acid sequence using the Kyte-Doolittle scale, with larger positive numbers indicating a more hydrophobic sequence⁴⁶."

It's written that Vamp2 shows 'no clear up and down pattern'. There is a pattern correlating increased GRAVY score with decreased elution/input however.

Thanks for pointing this out, we made this change.

The statement "Having determined the biophysical parameters that govern substrate binding to the Sti1domains" on p.10 is an overstatement and should be either made more specific or simply deleted.

We changed this to "Having identified biophysical features the define optimal Ubiquilin substrates"

Why is AlphaFold2 being used instead of AlphaFold3? A better program for disordered sequences might be IDPConformerGenerator.

We updated the figure with the AlphaFold 3 model.

We appreciate the suggestion to use IDPConformerGenerator, but we would like to point out that there are no predicted IDPs in our crystal structure, so we believe that AlphaFold 3 modeling is sufficient.

Labels need to be included on Figure 5A indicating Sti1-1, etc.

Thanks for this suggestion. We have added labels to identify the different STI domains.

Figure 5C: UBQLN2 needs to be immunoprobed. Is this assay done with the full length protein?

We apologize for the lack of clarity in our original manuscript. As described above, the same amount of recombinantly purified, full-length UBQLN2 is added to each experiment. We have also added Figure S4B to show that there is no difference in the IP efficiency of the different UBQLN2 mutants. We have clarified this in our revised manuscript.

This interpretation is not strictly correct "indicating that Sti1 domain interaction with placeholder sequences is conserved from yeast to humans (Figure 5C)." The loss of binding to P189T, which is in the Sti1 domain, written about later provides evidence for this interpretation. Also it would seem that binding to P189T is not statistically greater than background, making 'partial loss' not quite correct. Statistical analyses should be done for these data. If PH1 can bind to Sti1-2, then why is all binding lost for P189T?

This is a fair point. We have moved our discussion of the conservation of the STI1-placeholder interaction until after we show that the P189T mutation disrupts placeholder binding.

We performed additional replicates of the binding assay with PH1 and observe that the P440L mutation does have a statistically significant partial loss of binding compared to WT UBQLN2 (Figure 5D). While there is some variability, the P189T mutant does exhibit stronger binding than background.

It's not clear what the authors mean by "examination of the Agadir and GRAVY scores shows biophysical parameters consistent with the other placeholder sequences identified above (Figure 6A)." The GRAVY and Agadir scores in Figure 6A do not indicate consistency with Figure 4D.

This is a great point that we did not adequately address in the initial manuscript. Because the placeholder-STI1 interaction is intramolecular, it will have a high effective concentration. Therefore, the internal placeholders must have sub-optimal biophysical

properties for binding to the STI1 domains, otherwise no external substrate would be able to out-compete the intramolecular interactions. We have clarified this in the revised manuscript.

More characterization should be done on UBQLN2 P189T, since it is the consistent non-binder from which binding to Sti1-1 is proposed.

We have performed SEC analysis of P189T mutant and show that it behaves identical to the WT construct (Figure S4A). This demonstrates that there is no gross structural defect with the P189T mutant. We further show that there is no defect in IP efficiency with any of the UBQLN2 mutants (Figure S4B).

As described above, additional characterization added during revision argues that both STI1 domains can bind to substrates, but the STI1-I domain has higher affinity for substrates than the STI1-II domain. Serendipitously, the concentration of substrates and/or UBQLN2 used in our initial assays allowed for binding almost exclusively to the STI1-I domain, leading to the outsized effect of the P189T mutant. Repeating the binding assay with higher concentrations of UBQLN2 (Figures 2H, S4C, & S4D) eliminated the binding defect with the P189T mutant. Conversely, combining the P189T mutant with mutations in the STI1-II domain led to a loss of substrate binding (Figure 2I).

There is no figure showing the 'methionine rich hydrophobic groove.

We recognize that our original figure was difficult to interpret. We have clarified this in 1D.

Dr. Matthew L Wohlever
University of Pittsburgh
Cell Biology
Pittsburgh, Pennsylvania 15261

16th Feb 2026

Re: EMBOJ-2025-121480R
ALS mutations disrupt self-association between the Ubiquilin ST11 domain and placeholder sequences

Dear Matt,

Thank you for submitting your revised manuscript to The EMBO Journal. It has now been re-reviewed by two of the original referees, who were both fully satisfied with the revisions. We shall therefore be happy to proceed with acceptance and publication of the study, as soon as the following editorial issues have also been addressed:

- Please make sure to indicate the corresponding author's email address on the title page of the manuscript
- As we are switching from a free-text author contribution statement towards a more formal statement based on Contributor Role Taxonomy (CRediT) terms, please remove the present Author Contribution section and instead specify each author's contribution(s) directly in the Author Information page of our submission system during upload of the final manuscript. See <https://casrai.org/credit/> for more information.
- Please carefully go through the reference list, which contains many entries lacking full citation information (such as volume or page/eLocator numbers); please also update the citation to the originally cosubmitted study by Acharya et al
- In the Data Availability section, please include specific URLs for the two databases in which data generated during this work have been deposited and can be accessed.
- Please note that funding information as part of the Acknowledgements section should not have its separate header.
- Please double-check all figure references in the text, as some Figure panels (e.g. Fig 2C, Fig 5E-F) currently do not seem to be called out anywhere.
- Please move the Appendix Figure legends from the main text into the Appendix PDF, with each legend appearing directly underneath the respective figure. Please preface the Appendix title page with article information (E.g., 'Appendix for Onwunma et al, ALS mutations...'). Finally, make sure to reference the Appendix Figures always in full as 'Appendix Figure S1/2/3...' throughout the text.
- Please correct the labeling of the tables - the Reagents and Tools table should only be labeled as such, not as Table 2. Accordingly, Table 3 should be renamed as Table 2. Finally, please remove Table legends from the manuscript text, they should only appear in the respective Table files.
- Regarding included Source Data - please note that only items that have been provided should be checked in the Source Data Checklist. For the numerical Source Data of Figure 4, please make sure to clearly indicate which subpanel which part of the provided SD belong to. Finally, for those Source Data that have according to the checklist been externally deposited, please (a) make sure to spell this out explicitly (with URL and accession codes) in the Data Availability section (I assume this may be the Mendeley deposition, but this remains unclear).
- In our routine image screening, we noted that one blot panel in Figure 5C appears twice, despite being represented by distinct Source Data blots. Please carefully check the figure assembly and clarify/correct.
- Please provide suggestions for a short 'blurb' text prefacing and summing up the conceptual aspect of the study in two sentences (max. 250 characters), followed by 3-5 one-sentence 'bullet points' with brief factual statements of key results of the paper; they will form the basis of an editor-written 'Synopsis' accompanying the online version of the article. Please also upload a synopsis image, which can be used as a "visual title" for the synopsis section of your paper. The image (maybe based on Figure 7?) should be in JPG format, and please make sure that it remains in the modest dimensions of (exactly) 550 pixels wide and 300-600 pixels high.
- Finally, during pre-acceptance checks, our data editors have raised the following queries regarding figures, data, and legends; I would appreciate if you briefly answered to them in the cover letter of your final submission, and made the requested text

modifications with changes/additions highlighted via the "Track changes" option, to facilitate our final checking.

- 1) Please note that the exact p values are not provided in the legends of figures 2G, 5D-F; 6C
- 2) Please note that information related to n is missing in the legend of figure 2F
- 3) Please note that the measure of center for the error bars needs to be defined in the legends of figures 2G, 5D-F; 6C"

I am returning the manuscript to you for a final round of minor revision, solely to allow you to make these modifications and upload the revised files. Once we will have received them, we should be ready to swiftly proceed with formal acceptance and production of the manuscript.

With kind regards,

Hartmut

*** PLEASE NOTE: All revised manuscript are subject to initial checks for completeness and adherence to our formatting guidelines. Revisions may be returned to the authors and delayed in their editorial re-evaluation if they fail to comply to the following requirements. As a first step please read our guidelines for revised submissions:
<https://link.springer.com/journal/44318/submission-guidelines#cms-Revised-submissions>

1) Every manuscript requires a Data Availability section (even if only stating that no deposited datasets are included). Primary datasets or computer code produced in the current study have to be deposited in appropriate public repositories prior to resubmission, and reviewer access details provided in case that public access is not yet allowed.

4) Each main and each Expanded View (EV) figure should be uploaded as individual production-quality files (preferably in .eps, .tif, .jpg formats). For suggestions on figure preparation/layout, please refer to our Figure Preparation Guidelines:
<https://media.springernature.com/original/springer-cms/rest/v1/content/27825798/data/v1>

6) Please complete our Author Checklist, and make sure that information entered into the checklist is also reflected in the manuscript; the checklist will be available to readers as part of the Review Process File.

8) Please note that supplementary information at EMBO Press has been superseded by the 'Expanded View' for inclusion of additional figures, tables, movies or datasets; with up to five EV Figures being typeset and directly accessible in the HTML version of the article.

9) To facilitate reproducibility and cross-laboratory adoption of methodologies, please structure the Materials & Methods section as outlined in our guide to authors, including a completed Reagents and Tools Table.

10) Digital image enhancement is acceptable practice, as long as it accurately represents the original data and conforms to community standards. If a figure has been subjected to significant electronic manipulation, this must be clearly noted in the figure legend and/or the 'Materials and Methods' section. The editors reserve the right to request original versions of figures and the original images that were used to assemble the figure. Finally, we generally encourage uploading of numerical as well as gel/blot image source data.

In the interest of ensuring the conceptual advance provided by the work, we recommend submitting a revision within 3 months (17th May 2026). Please discuss the revision progress ahead of this time with the editor if you require more time to complete the revisions. Use the link below to submit your revision:

Link Not Available

Referee #1:

I have reviewed the authors response and revised manuscript and all of my concerns have been adequately addressed with some new control experiments, clarification of text/figures, and additional discussion.

Referee #2:

I am satisfied with the author responses.

UBQLNs are a family of proteins that play a key role in protein quality control by shuttling ubiquitinated proteins to the proteasome. UBQLNs consist of a UBL, UBA, and two ST11 domains. Mutations to the ST11 domains have been found in patients with amyotrophic lateral sclerosis (ALS). The authors solve the first structure of a ST11-domain from the Ubiquilin class of ST11 domain containing proteins, the Dsk2 ST11-domain bound to substrate. This structure reveals how the ST11 domain folds to form a hydrophobic groove and that it can dimerize and form a hydrophobic chamber, fully enclosing the hydrophobic segment of a substrate. They also demonstrated that ALS mutations disrupt the ST11-substrate binding, revealed that ST11 domain substrates are hydrophobic with low helical propensity, and identified internal regions within UBQLNs that bind to the ST11 domains as placeholder sequences to prevent the exposures of the ST11 domain hydrophobic groove.

Generally, the analyses are interesting and further our understanding of how the ST11-domain in ubiquilins preferentially bind to substrates. Furthermore, the authors identify internal regions within the UBQLNs that are capable of binding to the ST11 domain revealing how the domain remains folded and its hydrophobic groove protected in the absence of substrate.

In terms of minor comments:

"The Omp25 and Sec61b substrate series showed that Ubiquilins bound preferentially to TMDs with moderate hydrophobicity (Figure 3E-G & S3)" - the text currently says that E-F.

All minor editorial requests have been addressed by the authors.

Dr. Matthew L Wohlever
University of Pittsburgh
Cell Biology
Pittsburgh, Pennsylvania 15261

26th Feb 2026

Re: EMBOJ-2025-121480R1

ALS mutations disrupt self-association between the ubiquilin ST11 hydrophobic groove and internal placeholder sequences

Dear Dr. Wohlever,

Thank you for submitting your final revised manuscript for our consideration. I am pleased to inform you that we have now accepted it for publication in The EMBO Journal.

You may qualify for financial assistance for your publication charges - either via a Springer Nature fully open access agreement or an EMBO initiative. Check your eligibility: <https://link.springer.com/journal/44318/how-to-publish-with-us>

Yours sincerely,

Hartmut Vodermaier

Please note that it is The EMBO Journal policy for the transcript of the editorial process (containing referee reports and your response letters) to be published as an online supplement to each paper. If you should prefer removal of any referee-only figures included in the point-by-point response(s), e.g. because they may still be used for future publication or because they have been reproduced from published work by others, please do let us know immediately via response email.

More information is available here: <https://link.springer.com/partners/embo-press/editorial-policies#Peer%20review>